# ERNav: A Unified, Realistic Benchmark for Embodied AI with Exploration, Representation, and Navigation

## ABSTRACT

Current embodied AI benchmarks typically focus only on the final stage of the embodied process, such as following instructions or answering scene-related questions. These evaluations often unrealistically assume access to perfect perception data of the environment and overlook the earlier stages of exploration and representation construction, which are indispensable for real-world deployment. In addition, these benchmarks are often restricted to smaller-scale, room-level environments and short, object-centric instructions, falling to capture the complexity of larger buildings where agents must operate across multiple rooms and floors while reasoning over long instructions tied to global layouts. To address these gaps, we introduce ERNav, the first unified benchmark for embodied AI that integrates **E**xploration, **R**epresentation, and **Nav**igation into an end-to-end task pipeline. In ERNav, agents must actively explore the environment, construct global representations, and then localize targets directly from natural language instructions that often require reasoning over entire buildings. This unified formulation differs from existing benchmarks by aligning all stages of the embodied pipeline and scaling evaluation to realistic building-level settings, creating a challenging and practical testbed for embodied AI. We also propose 3D-LangNav as a strong baseline. As a divide-and-conquer framework, it employs a dual-sighted exploration strategy to collect diverse observations and construct high-quality 3D representations, followed by language grounding and spatial reasoning via a fine-tuned large language model (LLM). Extensive experiments show that ERNav poses significant new challenges for existing methods, while 3D-LangNav achieves strong performance, reaching more than twice the success rate (SR) of state-of-the-art 3D-MLLMs. Moreover, by structuring the task into three progressively harder, sequentially dependent subtasks as a whole pipeline, ERNav enables systematic analysis of how each stage contributes to overall performance, providing clear directions for future research.

## 1 INTRODUCTION

Embodied AI aims to develop agents that can perceive, act, and reason in realistic environments, enabling applications such as household assistance (Erickson et al., 2020) and robotics (Yuan et al., 2025). A long-standing challenge lies in evaluating such agents systematically in settings that reflect both the complexity of real-world environments and the interdependence among perception, representation, and reasoning. Recent advances in Large Language Models (LLMs) (Brown et al., 2020; Guo et al., 2025) and Multimodal LLMs (MLLMs) (Achiam et al., 2023; Bai et al., 2025) have driven progress in vision-language reasoning. However, their limitations in 3D perception and embodied tasks remain evident (Zha et al., 2025), underscoring the need for benchmarks that better capture the demands of embodied AI.

In response, several benchmarks (Ma et al., 2023; Achlioptas et al., 2020; Zhang et al., 2023) have been proposed for 3D scene understanding. However, two critical gaps remain. First, most benchmarks assume "free" access to complete RGB-D observations or ground-truth point clouds and evaluate only the final step—such as answering questions or grounding language in pre-scanned scenes. This design bypasses the earlier but essential stages of exploration and representation construction in real deployments. For example, SQA3D (Ma et al., 2023) provides complete scans and egocentric

videos upfront, making performance highly dependent on curated inputs rather than autonomous perception. Second, the environments in these benchmarks (e.g., ScanQA (Azuma et al., 2022), ScanRefer (Chen et al., 2020)) are typically restricted to single rooms or a few connected rooms (Zhi et al., 2025). Such settings limit instructions to short, object-centric references. In contrast, real-world scenarios involve multi-room and multi-floor buildings, where following instructions requires reasoning over long-range spatial contexts with complex layouts.

To address these gaps, we introduce **ERNav**, the first unified benchmark that integrates **E**xploration, **R**epresentation, and **Nav**igation into an end-to-end embodied task. ERNav reframes Vision-and-Language Navigation (VLN) (Anderson et al., 2018) from a robotics-inspired map-and-plan perspective (Durrant-Whyte & Bailey, 2006). Agents must first explore to construct a representation based on noisy RGB-D observations and then localize targets directly from natural language instructions. Unlike traditional navigation tasks, ERNav emphasizes destination identification, since low-level point-goal navigation has been extensively studied and is considered near-solved (Wijmans et al., 2019; Chaplot et al., 2020b). Instead, ERNav targets the practical challenge of interpreting language instructions while reasoning over large-scale environments. By combining diverse buildings from Matterport3D (Chang et al., 2017) with complex instructions from REVERIE (Qi et al., 2020), ERNav jointly evaluates active exploration and building-level spatial reasoning, two critical capabilities overlooked by existing benchmarks, as illustrated in Fig. 1.

To facilitate systematic analysis, we decompose ERNav into three interdependent subtasks: (1) **EnvExp**: efficient exploration for robust representation construction; (2) **EnvRep**: building global representations from exploration data (e.g., scene graphs or 3D language fields); (3) **EnvNav**: interpreting instructions over the constructed representation to identify the destination.

This modular, step-wise design is deliberately chosen over an end-to-end formulation to enhance interpretability and provide granular optimization targets. By isolating exploration, perception, and reasoning into separate stages, ERNav enables precise diagnosis of whether failures stem from insufficient coverage, flawed representation, or reasoning errors, rather than relying on a single binary success metric. This design philosophy is inspired by the step-by-step evaluation paradigm commonly adopted in Chain-of-Thought (CoT) reasoning, where intermediate steps are explicitly analyzed to better understand and improve model behavior. Moreover, this pipeline reflects real-world embodied systems that follow a "map-and-plan" paradigm, where a persistent global representation is built once and reused for subsequent tasks. In contrast to joint exploration–navigation approaches that must re-explore the environment for each new instruction, ERNav naturally supports a one-to-many evaluation setting, in which a single exploration pass can serve multiple navigation tasks, significantly reducing redundant computation and improving overall efficiency.

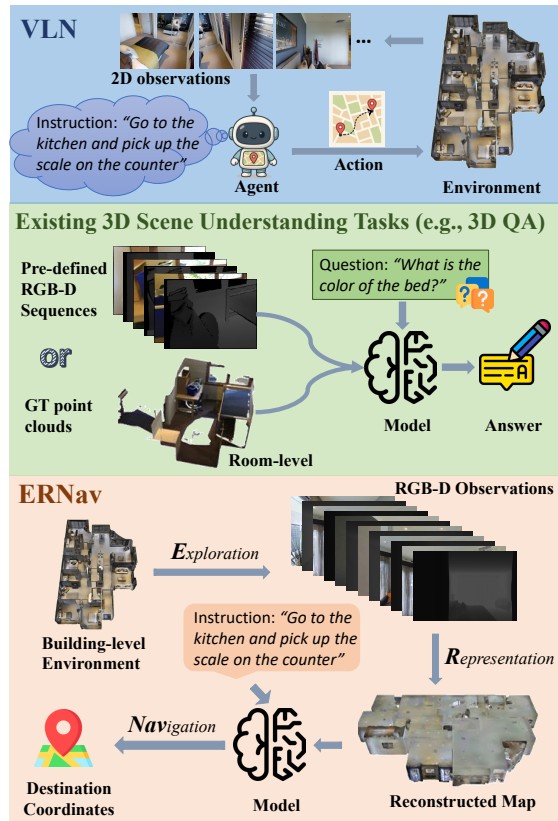

Figure 1: Comparison with VLN and existing 3D benchmarks. ERNav provides a realistic end-to-end setting: the agent actively explores a building-level environment, constructs a global representation, and grounds natural language instructions to directly predict the destination. The three subtasks—exploration, representation, and navigation—are interdependent, making ERNav a unified and realistic benchmark that extends beyond 2D-only navigation or room-level question answering.

One central challenge in ERNav lies in the scale of building-level environments. Processing large point clouds with thousands of objects demands both computational efficiency and strong generalization beyond room-scale datasets. To address this, we propose **3D-LangNav**, a divide-and-conquer baseline designed for ERNav. It employs a dual-sighted exploration strategy to improve reconstruction, builds object- and region-level representations through segmentation and graph aggregation, and leverages a fine-tuned LLM for spatial reasoning. This design reduces the reasoning problem to subgraphs centered on instruction-relevant landmarks, improving both efficiency and accuracy.

We benchmark a wide range of methods on ERNav and show that existing approaches fail to scale to building-level complexity. In contrast, 3D-LangNav achieves strong performance, attaining more than twice the success rate of state-of-the-art 3D-MLLMs. Importantly, it is the only method that unifies all three stages of ERNav while handling noise propagation across stages. Together, ERNav and 3D-LangNav establish a challenging and realistic testbed, and demonstrate the promise of structured exploration-mapping-reasoning pipelines for building-level embodied AI. In summary, our main contributions are:

- We introduce ERNav, the first unified benchmark that integrates embodied exploration, global 3D representation, and language-conditioned navigation in realistic building-scale environments, enabling the study of complete end-to-end embodied navigation workflows.

- We design novel task formulations and evaluation metrics for the three subtasks, including an object-centric coverage metric (ObjCov) for exploration, an auxiliary 3D representation evaluation for navigation, and a 3D scene understanding reformulation of VLN that, for the first time, enables 3D-MLLMs to perform language-grounded navigation.

- We propose 3D-LangNav, a technically novel end-to-end method with dual-sighted exploration, bi-level scene-graph matching, and LLM-based spatial reasoning, achieving strong performance across all ERNav subtasks and providing a strong baseline for future work.

- We conduct the first systematic cross-paradigm analysis of VLN agents, 3D representation methods, and 3D-MLLMs on a unified benchmark, revealing their complementary strengths and limitations and offering concrete insights for future embodied AI research in complex indoor environments.

## 2 RELATED WORK

**3D Scene Understanding Benchmarks.** A number of benchmarks (Ma et al., 2023; Achlioptas et al., 2020; Zhang et al., 2023) have been proposed for 3D scene understanding. Foundation datasets such as ScanNet (Dai et al., 2017) and Habitat-Matterport 3D (Ramakrishnan et al., 2021) provide large-scale RGB-D scans that support tasks including segmentation and detection (Kolodiazhnyi et al., 2024; 2025). Building on these, ScanRefer (Chen et al., 2020) enables 3D visual grounding from natural language, while ScanQA (Azuma et al., 2022) extends this to question answering. However, most benchmarks focus on small-scale, single-room settings rather than building-level reasoning. More recently, XR-Scene and XR-QA (Zhi et al., 2025) introduce cross-room reasoning, but remain limited to only a few connected rooms. A closely related benchmark is MSR3D (Linghu et al., 2024), which provides multimodal inputs and diverse QA-style tasks for 3D scene understanding and short-range navigation, primarily based on ScanNet environments. In contrast, ERNav operates in building-scale Matterport3D environments and targets long-horizon, language-conditioned navigation, requiring agents to reason over multi-room and multi-floor layouts. In addition, these benchmarks typically assume access to ground-truth scans or curated RGB-D sequences, bypassing the embodied challenges of exploration, noisy observations, and representation construction. ERNav departs from this paradigm by requiring agents to actively explore entire buildings, construct their own representations, and reason over multi-room, multi-floor layouts.

**Vision-and-Language Navigation (VLN).** Traditional VLN tasks such as R2R (Anderson et al., 2018) and SOON (Zhu et al., 2021) evaluate agents' ability to follow natural language instructions in photorealistic 3D environments. These tasks primarily assess step-by-step trajectory execution in previously unseen scenarios. Variants such as pre-explore settings (Wang et al., 2019) allow agents to survey the environment before navigation. However, they often rely on data augmentations such as back-translation (Wang et al., 2020) or panorama synthesis (Li & Bansal, 2024), and remain bound to discrete settings rather than continuous space (Krantz et al., 2020). Other extensions, including IVLN (Krantz et al., 2023) and GSA-VLN (Hong et al., 2025), adopt map-and-plan strategies with multiple trajectories in one environment, but their maps mainly support local decisions rather than

global reasoning. In contrast, ERNav reframes VLN as a 3D scene understanding problem. Agents explore once, build a global representation, and ground arbitrary instructions at the building level.

**Multimodal Scene Representations.** Recent work explores semantic-geometric representations of 3D environments by embedding objects and spatial relations within unified spaces. Early approaches (Zhang et al., 2022; Peng et al., 2023) align point clouds with CLIP features, while subsequent methods (Ding et al., 2023; Gu et al., 2024) incorporate relational and structural cues to support open-vocabulary queries. These representations facilitate tasks such as referring expression comprehension (Qiao et al., 2020), visual grounding (Roh et al., 2022), and open-vocabulary 3D understanding (Wu et al., 2024). Parallel efforts leverage multimodal LLMs (Xiong et al., 2025; Deng et al., 2025) for embodied tasks such as question answering (QA), dialogue, and planning, while more recent works (Kerr et al., 2023; Qin et al., 2024; Li et al., 2025) extend beyond point clouds by combining Neural Radiance Fields (Mildenhall et al., 2020) or 3D Gaussian Splatting (Kerbl et al., 2023) with vision-language features. For fair comparison, we focus on methods that directly process language queries, excluding those that require synthesizing RGB observations from camera poses, since this is impractical for embodied navigation. Nevertheless, most existing representations remain limited to object-centric reasoning, whereas ERNav evaluates long-range spatial reasoning over multiple landmarks—including both objects and regions—at the building scale.

## 3 THE ERNAV BENCHMARK

We now present the ERNav benchmark in detail, including the task formulation, and the definitions of its three subtasks.

### 3.1 TASK FORMULATION

ERNav differs from standard VLN by adopting a map-and-plan paradigm inspired by robotics: agents must first explore to construct a representation from RGB-D observations, and then localize the target directly from natural language instructions. Compared to the conventional setup where agents execute instructions step by step in an unseen environment, this formulation is both more realistic and more challenging. Formally, the agent is placed in an unseen environment $E$. Instead of immediately following instructions, it must perform a single exploration pass to collect RGB-D observations for representation construction. Let $T = \{\langle p_0, h_0 \rangle, \langle p_1, h_1 \rangle, \ldots, \langle p_N, h_N \rangle\}$ denote the exploration trajectory, where $p_i$ is the $i$-th position and $h_i$ its heading. Unlike VLN, ERNav assumes realistic egocentric observations $o_i$ rather than full panoramas $O$. From $T$, the agent builds a representation $R = f(T)$ which may take free form but must support open-vocabulary language queries. In the subsequent instruction-following stage, the agent is provided with an instruction $X$ and a starting point, and must directly predict the 3D coordinates $\hat{p}_t = (\hat{x}, \hat{y}, \hat{z})$ of the target location without further interaction with the environment. A prediction is considered correct if $\|\hat{p}_t - p_t\|_2 \leq 3\,\text{m}$, where $p_t$ is the ground-truth target coordinate.

### 3.2 SUBTASKS

To enable controlled evaluation across the pipeline, ERNav is decomposed into three subtasks.

#### 3.2.1 ENVEXP

In this subtask, the agent explores the environment once from a given starting point. We adopt Matterport3D (Chang et al., 2017) scenes with depth sensing restricted to $d_{max} = 10\,\text{m}$ following common practice (Chaplot et al., 2020a). To ensure fair comparison, we allow a sufficiently large step budget so that different exploration methods can fully cover the environment. Since Matterport3D contains multiple floors and cross-level exploration introduces unnecessary complexity, agents explore each floor independently. The final global trajectory is then obtained by connecting floors through the shortest paths across staircases.

**Metric.** Traditional exploration metrics emphasize map coverage, i.e., whether each region has been observed at least once. However, 3D reconstruction requires not only coverage but also multi-view observations of objects from diverse viewpoints and distances to capture fine details and reduce noise. Therefore, pure coverage fails to reflect the adequacy of exploration for reconstruction. To address this, we introduce a novel metric, ObjCov (object-wise distance coverage), which combines a coverage term $D$ and an efficiency term $E$. For each object $o \in \mathcal{O}$, we define its valid observation range as $[d_{\min}(o), d_{\max}(o)]$, where $d_{\min}(o)$ and $d_{\max}(o)$ denote the nearest and farthest observable

distances, with $o$ ($0 \leq d_{\min}(o) \leq d_{\max}(o) \leq d_{\max}$). We further define the effective distance range as $[L(o), U(o; \theta)]$, where $L(o) = d_{\min}(o)$ and $U(o; \theta) = \min(d_{\max}(o), \theta)$, with $\theta$ representing a reliability threshold for reconstruction. If $[\hat{d}_{\min}(o), \hat{d}_{\max}(o)]$ denotes the actual distance range from which $o$ is observed during exploration, the normalized distance coverage for $o$ is:

$$d(o; \theta) = \max\left( \frac{\min(\hat{d}_{\max}(o), U(o; \theta)) - \hat{d}_{\min}(o)}{U(o; \theta) - L(o)}, 0 \right). \tag{1}$$

The average object-wise coverage is then given by:

$$D(\theta) = \frac{1}{|\mathcal{O}|} \sum_{o \in \mathcal{O}} d(o; \theta) \tag{2}$$

To penalize redundant trajectories, we introduce an efficiency term $E = 1 - \sqrt{\frac{S_{\text{actual}}}{S_{\max}}}$, where $S_{\text{actual}}$ is the number of steps taken and $S_{\max}$ is the brute-force step count required to visit all positions. Since most objects can be reliably observed at 3 m, we set $\theta = 3$ and define the final metric as:

$$\text{ObjCov} = D(\theta) \times E \tag{3}$$

Besides these exploration-related metrics, EnvExp further assesses how exploration trajectories affect the quality of subsequent reconstruction. Specifically, we back-project all 2D pixels into the 3D map to build point clouds and compare them with the ground-truth reconstructions using metrics such as F1 score and the Chamfer-L1 distance.

### 3.2.2 ENVREP

Given the exploration observations $O = \langle o_1, o_2, \ldots, o_n \rangle$, the agent is required to construct a 3D representation $R = f(O)$ that supports natural language queries for retrieving 3D coordinates. To ensure comparability, we provide standardized trajectories generated by our 3D-LangNav baseline as well as ground-truth point clouds for evaluation. Unlike reconstruction-focused tasks, EnvRep emphasizes the utility of representations for navigation rather than visual fidelity. Therefore, instead of evaluating with traditional downstream tasks such as semantic segmentation, we design an auxiliary evaluation aligned with ERNav instructions. Specifically, we parse landmarks from the instructions and query the built representation $R$ to retrieve candidate positions for each landmark. We then measure performance using ranking-based metrics: Recall@k, which measures the fraction of cases where the ground-truth landmark is ranked within the top $k$; MR (mean rank), which computes the average position of the ground-truth in the ranking; and MRR (mean reciprocal rank), which evaluates the average reciprocal rank across cases. An effective representation achieves high Recall@k and MRR while maintaining low MR, thereby ensuring that navigation instructions can be accurately grounded to 3D coordinates.

### 3.2.3 ENVNAV

In EnvNav, the agent is given a starting position $(x_0, y_0, z_0)$ and a natural language instruction $W$ describing the target or its surrounding context. We adopt object-centric instructions from REVERIE (Qi et al., 2020), which reflect realistic navigation goals, and leave the inclusion of other tasks like R2R (Anderson et al., 2018) and SOON (Zhu et al., 2021) for future work. The agent must predict the destination coordinates using the instruction $W$ and the representation $R$ built in EnvRep, without any further interaction with the environment. Unlike retrieval or grounding tasks such as ScanRefer (Chen et al., 2020) that require exact object localization, EnvNav only demands identifying the vicinity of the target. This design follows the VLN convention, where fine-grained localization and manipulation can be deferred to 2D image-based methods, which are more robust to real-world dynamics (e.g., objects being moved slightly after exploration). For evaluation, we adopt two standard VLN metrics: (1) Success Rate (SR): the proportion of predictions that fall within 3 m of the ground-truth destination. (2) Navigation Error (NE): the Euclidean distance between the predicted location and the target.

### 3.3 DATASET STATISTICS

ERNav is constructed by reorganizing and integrating several well-established datasets, rather than collecting new raw data from scratch. All data used in the benchmark are therefore inherited from

Figure 2: Overview of 3D-LangNav, including the dual-sighted exploration strategy for EnvExp, hierarchical mapping for EnvRep, and LLM-based spatial reasoning for EnvNav.

widely adopted, carefully curated sources, whose quality and reliability have been extensively validated in prior work.

**EnvExp.** The EnvExp subtask is built upon the validation and test splits of Matterport3D. We perform floor-level exploration over 29 scenes, covering a total of 61 floors. For each floor, three starting positions are randomly sampled, resulting in 183 exploration episodes in total.

**EnvRep and EnvNav.** Both EnvRep and EnvNav are built upon the unseen validation split of REVERIE, which contains 10 large-scale indoor scenes and 3,573 trajectory–instruction pairs. This split is directly used for the navigation evaluation in EnvNav. However, because the unseen scenes contain relatively few region-level landmarks, we additionally incorporate region-level landmarks from the REVERIE training split to ensure sufficient semantic coverage for representation evaluation. As a result, EnvRep consists of 3,297 object-level landmarks and 1,324 region-level landmarks, all sourced from REVERIE's official annotations. All data used in ERNav are therefore derived from rigorously validated benchmarks, ensuring both data quality and representativeness.

# 4  3D-LANGNAV

We now describe 3D-LangNav, a strong baseline that adopts a divide-and-conquer strategy for ER-Nav, as shown in Fig. 2.

## 4.1  METHOD OVERVIEW

We formulate instruction-following navigation as a scene-graph matching problem, where the instruction is represented as a subgraph and the environment as a graph of landmarks and spatial relations. Landmarks may correspond to objects (e.g., a table) or regions (e.g., a bedroom), while edges encode spatial relations such as adjacency or containment. A key challenge is that region-level landmarks are inherently ambiguous, since rooms often lack clear boundaries. Prior work, such as HOV-SG (Werby et al., 2024), relies on rule-based segmentation, but this approach introduces brittle assumptions, tightly couples performance to segmentation quality, and ignores linguistic subjectivity. For example, a kitchen and living room without a dividing wall may be perceived as a single region or two distinct ones. Thus, instead of requiring exact room inference, we aim to construct representations that allow robust grounding of both object- and region-level landmarks, while keeping the subsequent matching process computationally efficient.

## 4.2 DUAL-SIGHTED EXPLORATION

Standard exploration methods prioritize coverage but often fail to balance detailed local views and global contextual observations, producing reconstructions that are both sparse and noisy. Consequently, existing 3D scene understanding approaches frequently rely on human-collected trajectories, which are costly and biased (see Appendix for visualizations of exploration trajectories).

We propose a training-free dual-sighted exploration strategy that can be seamlessly integrated into existing exploration methods. The agent is equipped with two complementary "eyes". The first is a near-sighted eye with perception limited to $d_{near}$ meters for dense local observations, and the other is a far-sighted eye, which is restricted to beyond $d_{far}$ for capturing global spatial relations. During exploration, the agent maintains two frontier sets $(p_{\text{near}}, p_{\text{far}})$ and updates two coverage maps in parallel. By default, the agent prioritizes $p_{\text{near}}$ except when (i) no near frontiers remain, or (ii) $p_{\text{far}}$ lies in a region already covered locally but not from a distance. This allows near-sighted exploration to ensure dense local coverage, while far-sighted exploration complements it with contextual cues. Note that although perception is separated, all recorded observations still retain the maximum depth $d_{\max}$. The strategy produces one-pass trajectories that ensure dense local coverage while preserving global context, reducing redundancy, and supporting high-quality 3D reconstruction.

## 4.3 3D REPRESENTATION CONSTRUCTION

We then build a hierarchical representation of the environment. For objects, we follow HOV-SG (Werby et al., 2024) to first use a class-agnostic segmentation model (Kirillov et al., 2023) to segment each frame and extract its CLIP embedding. These segments are then projected into 3D, merged across frames, and denoised with DBSCAN (Ester et al., 1996) to construct the object map. For regions, we argue that exact room inference is unnecessary for navigation. Instead, the agent only needs to verify whether a sub-region belongs to the described room type. To this end, we create a region map using the Voronoi algorithm applied to the viewpoints bypassed during exploration. These points cover the entire environment while maintaining sufficient spacing, such that each point effectively represents a spatially coherent area. Each node is assigned the mean CLIP features aggregated from nearby RGB observations. This design avoids brittle rule-based segmentation, mitigates ambiguity, and ensures full spatial coverage.

## 4.4 BI-LEVEL SCENE-GRAPH MATCHING

We reduce ERNav to finding the best-matching subgraph between the instruction graph and the environment graph. Previous global matching approaches suffer from combinatorial complexity, while attention-based methods such as LSceneLLM (Zhi et al., 2025) require bi-level optimization. In contrast, 3D-LangNav adopts a node-first, edge-verification strategy: candidate landmarks are matched first, and spatial relations are verified afterwards. This substantially reduces search complexity while maintaining robust alignment (see Appendix for analysis).

We first perform hierarchical queries utilizing the 3D representations constructed. For object- and region-level landmarks, we compute similarity scores between text and visual embeddings and then apply both thresholding $\alpha$ and top-$k$ selection to filter candidates for both easy queries like "table" and hard ones like "leopard decoration". For higher-level queries such as "the first floor", we defer reasoning to the spatial reasoning module as it requires no visual information. Given candidate landmarks, we then resolve their spatial relations according to the instructions.

Existing methods either (i) verify relations pairwise with MLLMs on 2D images, or (ii) encode all possible relations into large scene graphs. Both approaches are inefficient and not suitable for long-range reasoning. We instead leverage the reasoning capability of LLMs to perform spatial grounding in a single step. Specifically, we perform parameter-efficient finetuning (PEFT) on a powerful LLM, Qwen2.5-72B (Team, 2024). Training data are generated from ground-truth segmentations and OpenScene (Peng et al., 2023) features from the train split of REVERIE. Prompts are designed to include five parts: the instruction, starting coordinates, candidate landmarks, the target, and navigable points for floor/structural reasoning. To enhance robustness, we augment data by perturbing coordinates and varying candidate set sizes. The fine-tuned LLM directly predicts the final target coordinates, avoiding multi-stage optimization and enabling efficient, robust spatial reasoning.

Table 1: Exploration and reconstruction metrics of different methods in EnvExp.

| Methods | Exploration | | | Reconstruction | | | |
|---|---|---|---|---|---|---|---|
| | Coverage%↑ | $M(3)$%↑ | $ObjCov$%↑ | F1%↑ | Precision%↑ | Recall%↑ | Chamfer-L1 (m)↓ |
| FBE | 93.9 | 33.1 | 27.2 | 45.4 | 69.8 | 33.7 | 0.507 |
| ANS | 91.5 | 35.8 | 29.0 | 47.7 | 70.3 | 36.1 | 0.428 |
| PONI | 88.0 | 33.6 | 27.0 | 39.6 | 67.8 | 27.9 | 0.589 |
| LFE | 92.2 | 34.7 | 27.9 | 43.7 | 69.2 | 31.9 | 0.516 |
| Human | 96.2 | 31.1 | 23.2 | 45.7 | 70.4 | 33.6 | 0.569 |
| ANS+DSS | **99.4** | **60.7** | **41.0** | **52.9** | **71.2** | **42.4** | **0.343** |

Table 2: Comparison of different 3D representation-based methods in EnvRep.

| Methods | Seg. | Object-level | | | | | Region-level | | | | |
|---|---|---|---|---|---|---|---|---|---|---|---|
| | | R@1%↑ | R@10%↑ | R@20%↑ | MR↓ | MRR%↑ | R@1%↑ | R@10%↑ | R@20%↑ | MR↓ | MRR%↑ |
| OpenScenes-2D | Pred. | 23.4 | 61.4 | 77.0 | 18.6 | 35.8 | 23.3 | 49.8 | 63.8 | 35.0 | 32.4 |
| | GT | 26.2 | 74.4 | 90.0 | 8.6 | 41.2 | 18.5 | 67.6 | 88.9 | 11.8 | 34.7 |
| OpenScenes-3D | Pred. | 10.4 | 37.0 | 52.2 | 50.9 | 18.8 | 10.8 | 35.5 | 50.2 | 53.6 | 19.6 |
| | GT | 9.5 | 48.8 | 69.1 | 21.4 | 21.6 | 14.3 | 54.0 | 65.9 | 19.0 | 25.2 |
| OpenScenes-Ens | Pred. | 10.4 | 44.5 | 57.9 | 32.1 | 21.5 | 9.1 | 49.1 | 63.4 | 23.6 | 20.6 |
| | GT | 13.6 | 56.8 | 74.5 | 18.2 | 26.6 | 9.4 | 57.1 | 79.1 | 13.0 | 26.2 |
| ConceptGraphs | Pred. | 18.5 | 59.4 | 73.3 | 27.5 | 31.2 | 18.1 | 71.4 | 87.5 | 14.6 | 32.9 |
| HOV-SG | Pred. | 17.5 | 69.9 | 84.2 | 12.1 | 34.5 | 12.2 | 68.9 | 75.0 | 21.5 | 32.5 |
| 3D-LangNav | Pred. | 17.5 | 69.9 | 84.2 | 12.1 | 34.5 | 35.2 | 78.7 | 88.9 | 8.0 | 50.3 |

## 5 EXPERIMENTS

### 5.1 EXPERIMENTAL SETUP

**Baseline Methods.** For **EnvExp**, we evaluate representative exploration methods, including FBE (Yamauchi, 1997), ANS (Chaplot et al., 2020a), PONI (Ramakrishnan et al., 2022), and LFE (Li et al., 2023). We also include human demonstrations, partly from VLMaps (Huang et al., 2023) and partly manually collected. For **EnvRep**, we benchmark methods capable of constructing 3D representations from RGB-D observations or point clouds. These include three variants of OpenScenes (Peng et al., 2023), ConceptGraphs (Gu et al., 2024), and HOV-SG (Werby et al., 2024). For **EnvNav**, we consider three categories of methods: (i) VLN models, (ii) representation-based methods, and (iii) 3D-MLLMs. Details of these baselines are provided in the Appendix.

**Implementation Details.** In **EnvExp**, each method is run from three different starting positions, and the trajectory with the largest coverage is retained. We set $d_{near} = 1\,\text{m}$ and $d_{far} = 3\,\text{m}$ for the 3D-LangNav exploration. The instructions and environments are from the val unseen split of REVERIE. For region-level landmarks, we additionally include data from the train split for EnvRep evaluation. For 3D-MLLMs, we adapt their preprocessing strategies to Matterport3D for fair comparison. For 3D-LangNav, candidate proposals are generated with $\alpha = 0.25$ and $k = 25$. LoRA finetuning uses rank $r = 8$ across all layers, learning rate $1 \times 10^{-4}$, and is implemented with *Llama-Factory* (Zheng et al., 2024) on 16 NVIDIA H100 GPUs. More implementation details for the baselines can be found in the appendix.

### 5.2 MAIN RESULTS

#### 5.2.1 ENVEXP

Tab. 1 reports the results of our Dual-Sighted exploration Strategy (DSS) against baselines, evaluated on both exploration and reconstruction metrics. For exploration metrics, DSS significantly improves both traditional coverage and the proposed $M(3)$ and ObjCov, demonstrating its ability to produce more diverse and efficient object observations. For reconstruction metrics, DSS consistently outperforms baselines across F1, precision, recall, and Chamfer-L1 distance, confirming that our metrics reliably correlate with downstream reconstruction quality. Although DSS typically requires more exploration, this additional cost is already explicitly penalized in our ObjCov metric, and the superior performance of DSS indicates that the gain in multi-view observation quality outweighs the efficiency trade-off. Moreover, because ERNav targets 3D reconstruction rather than 2D free-space discovery, additional viewpoints are physically necessary, making fixed-step comparisons both technically infeasible and misaligned with real-world "map-and-plan" workflows. Together, these findings highlight two conclusions: (1) DSS enables more effective exploration than prior methods, and (2) The proposed metrics capture exploration quality beyond simple coverage.

### 5.2.2 ENVREP

We evaluate the grounding performance of different 3D representations using ERNav instructions. To evaluate with different types of queries, we classify landmarks into object-level and region-level categories using GPT-4 (Achiam et al., 2023) and report their results separately in Tab. 2. Since OpenScenes only provides point-wise features without object segmentation, we evaluate it under two conditions: (1) using ground-truth segmentation, and (2) using DBSCAN-based segmentation to cluster points into objects, as in prior work (Huang et al., 2025b).

The results reveal two key findings. First, better segmentation improves object-level grounding, but has little effect on region-level ones. This is intuitive as object features are computed by averaging point cloud features, making a more accurate segmentation produces better-aligned and less noisy features. In contrast, many region-level landmarks (e.g., "hallway") cannot be inferred from the presence of individual objects, since such objects are often common to multiple regions. Second, despite sharing the same object-level pipeline as HOV-SG, 3D-LangNav achieves substantially higher region-level performance. This improvement benefits from our design of using navigable nodes and a hierarchical query strategy, which aggregates contextual features without relying on brittle room segmentation. These results highlight the importance of robust region-level reasoning and demonstrate that 3D-LangNav better bridges language with 3D scene representations. Although 3D-LangNav achieves a slightly lower R@1% compared to point-based methods, this reflects a deliberate trade-off between fine-grained texture precision and object-level robustness through feature aggregation. More importantly, for navigation, improving Recall@k and region-level grounding is more critical than optimizing Top-1 precision, and 3D-LangNav consistently outperforms baselines on both, which directly contributes to its superior downstream navigation performance.

### 5.2.3 ENVNAV

We evaluate three categories of methods on EnvNav, with results in Tab. 3. Strictly speaking, existing VLN methods are not directly comparable, as they omit exploration and representation and instead explore interactively during inference. Nevertheless, we include them for comparison since they share the same ultimate goal of localizing a destination from natural language instructions.

Table 3: Comparison of three types of methods in EnvNav. Each category highlights subtask coverage. 3D-LangNav is the only method that covers all three subtasks, enabling an end-to-end solution.

| Methods | NE↓ | SR↑ |
|---|---|---|
| *VLN methods*
Exp:✗  Rep:✗  Nav:✓ | | |
| VLNBERT (Hong et al., 2021) | 5.74 | 44 |
| CM² (Georgakis et al., 2022) | 7.02 | 34 |
| DUET (Chen et al., 2022) | 5.14 | 37 |
| GridMM (Wang et al., 2023) | 4.21 | 49 |
| InstructNav (Long et al., 2024) | 6.89 | 31 |
| NaVid (Zhang et al., 2024b) | 5.47 | 37 |
| Uni-NaVid (Zhang et al., 2024a) | 5.58 | 47 |
| RAM (Wei et al., 2025) | 4.95 | 44 |
| COSMO (Zhang et al., 2025) | - | 47 |
| Dynam3D (Wang et al., 2025) | 5.34 | 53 |
| *Representation-Based Methods*
Exp:✗  Rep:✓  Nav:✓ | | |
| OpenScenes-2D (Peng et al., 2023) | 8.80 | 23 |
| OpenScenes-3D (Peng et al., 2023) | 9.71 | 10 |
| OpenScenes-Ens (Peng et al., 2023) | 9.04 | 14 |
| ConceptGraphs (Gu et al., 2024) | 10.01 | 17 |
| HOV-SG (Map) (Werby et al., 2024) | 10.59 | 15 |
| HOV-SG (Nav) (Werby et al., 2024) | 9.54 | 27 |
| *3D-MLLMs*
Exp:✗  Rep:✓  Nav:✓ | | |
| ChatScene (Huang et al., 2024) | 8.31 | 15 |
| Reason3D (Huang et al., 2025b) | 10.55 | 8 |
| LSceneLLM (Zhi et al., 2025) | 11.03 | 5 |
| 3D-LLaVA (Deng et al., 2025) | 8.96 | 13 |
| LLaVA-3D (Zhu et al., 2025) | 7.80 | 18 |
| Human | - | 81 |
| 3D-LangNav (Exp:✓  Rep:✓  Nav:✓) | 5.15 | 50 |

A key distinction revealed in Tab. 3 is the subtask coverage. VLN methods address only navigation, while representation-based methods and 3D-MLLMs incorporate instruction reasoning over 3D representations but bypass exploration. 3D-LangNav is the only approach covering all three subtasks, enabling a fully end-to-end solution. This makes the setting more realistic but also more challenging, as errors and noise from exploration and representation propagate into navigation. Despite this, 3D-LangNav achieves results on par with the strongest VLN methods (e.g., Dynam3D) and substantially outperforms all representation-based and 3D-MLLM baselines, while requiring only one-shot

inference without environment interaction. Beyond competitive SR, the proposed map-and-plan paradigm provides additional structural advantages over step-wise VLN agents, including reusable global maps for one-to-many instructions, optimal path execution via classical planners, and explicit interpretability across exploration, representation, and reasoning stages. While coordinate-based prediction may incur higher NE than continuous path-following agents, it directly reflects the difficulty of global target localization and is more aligned with real-world deployment, where identifying the correct destination is more critical than stopping near it. These results demonstrate that 3D-LangNav is not only competitive with strong VLN baselines, but also establishes a scalable and interpretable alternative for building-scale language-conditioned navigation.

Representation-based methods perform close to their R@1 scores in Tab. 2, reflecting their grounding-oriented design and limited ability to interpret complex navigation instructions. HOV-SG improves performance by hierarchically parsing instructions into floors, rooms, and objects, but assumes single-entity references and perfect room segmentation. By removing these constraints, 3D-LangNav nearly doubles the SR of HOV-SG.

Another important finding is that current 3D-MLLMs perform poorly on EnvNav, with SR consistently below 20%. We attribute this to two factors. First, EnvNav involves significantly larger point clouds and more complex instructions, exceeding the capacity of models trained on room-level scenes with localized references. Second, most 3D-MLLMs are trained with QA-style supervision, which brings strong semantic understanding but weak coordinate prediction. For example, ChatScene (Huang et al., 2024) relies only on implicit local spatial relations from DINOv2 (Oquab et al., 2023) features without explicit relational encoding. While effective on ScanRefer, this design fails on EnvNav, which requires long-range reasoning and explicit coordinate awareness. 3D-LangNav addresses these challenges by combining structured representation with LLM-based reasoning, reformulating open-ended coordinate prediction as a multi-choice problem. Although implemented as a two-stage baseline, it is the first to span exploration, representation, and navigation, demonstrating the feasibility of an end-to-end embodied solution.

### 5.3 ABLATION STUDY

**Different LLMs.** Table 4 compares 3D-LangNav in EnvNav using different LLM backbones. For Llama3.1-8B (Dubey et al., 2024) and Qwen2.5-7B (Team, 2024), we apply full finetuning, as this generally yields stronger performance. For Qwen2.5-72B, we adopt LoRA finetuning due to the prohibitive cost of full finetuning. The results show that model capacity is the dominant factor in performance, as the base Qwen2.5-72B significantly outperforms fully finetuned 7B and 8B models in both

Table 4: Comparison of 3D-LangNav with different LLMs in EnvNav.

| LLM | Train | NE↓ | SR↑ |
|---|---|---|---|
| Llama3.1-8B | ✗ | 6.35 | 32 |
| | ✓ | 6.09 | 35 |
| Qwen2.5-7B | ✗ | 6.16 | 35 |
| | ✓ | 6.02 | 37 |
| Qwen2.5-72B | ✗ | 5.69 | 42 |
| | ✓ | **5.15** | **50** |

metrics. The 72B model also shows the greatest potential for improvement, achieving the largest SR gain (+8%) even with LoRA. These findings suggest that advances in LLMs will further enhance the performance of 3D-LangNav.

**Impact of Data Augmentation.** We also study the effect of different augmentation strategies during LLM finetuning, summarized in Tab. 5. Three augmentations are considered: (1) Candidate Shuffle (**CS**), randomly shuffling landmark orders and identifiers; (2) Candidate Varying (**CV**), altering $\alpha$ and $k$ to vary the size of candidate sets; (3) Position Bias (**PB**), adding random offsets to candidate coordinates. Additionally, we evaluate a variant that removes the navigable node information in #1. Each augmentation brings measurable

Table 5: Ablation of augmentation strategies for 3D-LangNav in EnvNav.

| # | Augmentation | NE↓ | SR↑ |
|---|---|---|---|
| 0 | Base (no aug.) | 5.33 | 47.8 |
| 1 | w/o navigable nodes | 5.50 | 45.8 |
| 2 | Candidate shuffle (CS) | 5.25 | 48.5 |
| 3 | Candidate varying (CV) | 5.28 | 48.7 |
| 4 | Position bias (PB) | 5.22 | 49.4 |
| Ours | **All (CS+CV+PB)** | **5.15** | **49.9** |

gains. The navigable nodes prove particularly important, as they provide structural and level information critical for grounding instructions. CS improves robustness to variable candidate sets and mitigates dataset-specific biases. CV exposes the model to a broader range of candidate distribu-

tions, while PB enforces reasoning over relative spatial relations rather than memorizing absolute coordinates. Finally, combining all augmentations achieves the best performance.

## 6 CONCLUSION

In this paper, we present ERNav, a novel benchmark for building-level scene understanding in embodied navigation. By interpreting VLN from a scene understanding perspective, ERNav introduces three complementary subtasks—environment exploration, map construction, and scene comprehension—that together establish a realistic, end-to-end evaluation pipeline for real-world navigation. To accompany this benchmark, we proposed 3D-LangNav, a strong baseline that combines a dual-sighted exploration strategy with a two-stage reasoning framework: generating landmark candidates followed by LLM-based spatial reasoning. Through extensive experiments across all subtasks, we show that ERNav enables systematic evaluation of embodied navigation methods, while 3D-LangNav consistently outperforms strong baselines and highlights the importance of unified solutions. We hope this work will inspire further research into more generalizable and scalable approaches for embodied navigation in complex 3D environments.

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

# A  APPENDIX

This document provides additional method details, supplementary experiments, and further analysis to complement the main paper, including:

- Appendix A.1: detailed descriptions of the baseline methods.
- Appendix A.2: implementation details of the baseline methods.
- Appendix A.3: more experimental results on EnvExp.
- Appendix A.4: ablation study on different camera configurations.
- Appendix A.5: experiments on the influence of early stages on final navigation.
- Appendix A.6: complexitity analysis of our 3D-LangNav.
- Appendix A.7: discussions the real-world applications of applying ERNav.
- Appendix A.8: visualizations of exploration trajectories from different methods.
- Appendix A.9: prompt templates used in 3D-LangNav.
- Appendix A.10: how 3D-LangNav deal with dynaic scenes.
- Appendix A.11: explanations of the noises in ERNav.
- Appendix A.12: the use of LLMs in this work.
- Appendix A.13: discussions on the limitations and future directions of this work.

## A.1  BASELINE METHODS

In this section, we provide detailed descriptions of the baseline methods used in our experiments.

### A.1.1  ENVEXP BASELINES

**Frontier-based Exploration (FBE) (Yamauchi, 1997)**  is a classic heuristic strategy that guides the agent toward the closest unexplored frontier at each step. Without relying on learning, it greedily expands coverage by incrementally pushing the boundary of known space. This simple rule-based approach serves as a strong traditional baseline for evaluating exploration methods.

**Active Neural SLAM (ANS) (Chaplot et al., 2020a)**  combines deep reinforcement learning with classical frontier-based planning. A global policy, trained to maximize coverage, proposes long-term goals from the agent's occupancy map and visitation history, while the nearest frontier to that goal is chosen for navigation to improve robustness. This hybrid design allows the method to leverage learned strategies while retaining stability from rule-based exploration.

**PONI (Ramakrishnan et al., 2022)**  builds on frontier-based exploration by ranking frontiers according to their spatial and semantic potential. In this simplified variant, only the learned area estimation from the UNet is used, and the agent consistently selects the frontier with the largest predicted coverage. This results in a purely greedy but informed exploration strategy.

**Learning-Augmented Model-Based Frontier-Based Exploration (LFE) (Li et al., 2023)**  tackles exploration under strict time limits by combining learning with model-based planning. It predicts both the unexplored area behind each frontier and the steps needed to reach it, allowing the planner to balance efficiency and completeness. By integrating semantic cues and structured decision-making, it improves over purely greedy or RL-based strategies in coverage.

### A.1.2  ENVREP BASELINES

**OpenScenes (Peng et al., 2023)**  introduces a zero-shot framework for open-vocabulary 3D scene understanding by aligning 3D points with both text and image features in the CLIP space. It extracts three types of features: a 2D branch, where multi-view pixel features are fused after back-projecting points into posed images; a 3D branch, where sparse convolutions capture geometric structure directly from the point cloud; and an ensemble branch, which combines the two for richer, more robust representations. This hybrid design allows OpenScenes to ground natural language queries in 3D scenes with greater accuracy and flexibility than either 2D or 3D features alone.

**ConceptGraphs (Gu et al., 2024)**   proposes a graph-structured representation of 3D scenes that moves beyond dense per-point features. By leveraging 2D foundation models and fusing their outputs into 3D through multi-view association, it builds compact graphs where nodes capture semantic entities and edges encode their spatial relationships. This design not only supports open-vocabulary generalization to novel classes but also enables downstream planning tasks that demand higher-level reasoning over both spatial and semantic concepts.

**HOV-SG (Werby et al., 2024)**   introduces a hierarchical open-vocabulary scene graph for 3D mapping and navigation. Instead of relying on dense per-point features, it organizes environments into a multi-level structure of floors, rooms, and objects, each enriched with language-aligned representations from vision foundation models. This hierarchy makes large and complex spaces more manageable, enabling efficient cross-floor navigation and stronger performance in language-conditioned tasks while keeping the representation compact.

### A.1.3   EnvNav Baselines: VLN

**VLNBERT (Hong et al., 2021)**   adapts the transformer architecture to the navigation setting by introducing a recurrent mechanism that preserves cross-modal state over time. This design allows the model to handle partially observable environments while aligning language instructions with visual inputs. It simplifies the navigation pipeline, achieving strong results without the need for more complex encoder–decoder structures.

**CM$^2$ (Georgakis et al., 2022)**   takes a different angle on VLN by grounding language directly into spatial maps rather than relying solely on sequence models or raw attention over observations. It learns to infer semantic top-down maps, even for unseen areas, and then plans navigation paths as waypoints guided by language. This explicit map-based reasoning leads to more structured and interpretable navigation compared to purely end-to-end approaches.

**DUET (Chen et al., 2022)**   introduces a dual-scale graph transformer that balances local grounding with global planning in VLN. It constructs a topological map during navigation for efficient long-term reasoning, while also attending to fine-grained visual–language alignment through local observations. By combining these two levels of representation, DUET achieves strong performance across both coarse and fine-grained navigation tasks.

**GridMM (Wang et al., 2023)**   tackles VLN by introducing a grid-based memory map that grows dynamically as the agent explores. It projects past observations into a unified top-down grid to capture spatial structure, while also aggregating instruction-relevant details within each grid cell. This combination of global spatial reasoning and local language grounding leads to strong navigation performance across multiple benchmarks.

**InstructNav (Long et al., 2024)**   is designed as a general-purpose system for handling diverse navigation instructions without relying on task-specific training or pre-built maps. It introduces a Dynamic Chain-of-Navigation (DCoN) to unify planning across different instruction types and leverages Multi-sourced Value Maps to translate language into executable trajectories. This flexible design enables strong zero-shot performance across multiple navigation tasks, including real-world robot experiments.

**NaVid (Zhang et al., 2024b)**   introduces a video-based large vision-language model that performs navigation directly from RGB streams, without relying on maps, odometry, or depth inputs. By treating navigation as spatio-temporal reasoning over continuous video and language instructions, it mimics how humans navigate and avoids common Sim2Real issues. This design enables NaVid to achieve strong performance in both simulated and real-world environments, highlighting the potential of VLMs for robust instruction-following navigation.

**Uni-NaVid (Zhang et al., 2024a)**   extends the idea of video-based navigation to a broader scope, aiming to unify diverse embodied tasks such as instruction following, object search, and human tracking within a single model. By standardizing inputs and outputs across tasks, it enables a generalist agent that can seamlessly handle mixed long-horizon navigation demands in unseen environ-

ments. Trained on millions of samples from multiple subtasks, Uni-NaVid demonstrates both strong benchmark performance and practical effectiveness in real-world trials.

**RAM (Wei et al., 2025)** tackles the persistent problem of data scarcity in VLN by generating fresh observation–instruction pairs through rewriting rather than collecting new simulator or web data. It enriches observations with diverse objects and layouts using VLM–LLM rewriting plus text-to-image synthesis, and then produces aligned instructions by reasoning over the differences from the originals. Combined with a tailored training strategy, this approach diversifies the data distribution while keeping noise in check, leading to stronger generalization across both discrete and continuous VLN benchmarks.

**COSMO (Zhang et al., 2025)** aims to strike a balance between performance and efficiency in VLN, addressing the rising complexity of transformer-based methods that often depend on external knowledge or maps. It combines state-space and transformer modules, introducing two tailored components: RSS for stronger inter-modal interactions and CS3 for dual-stream cross-modal reasoning. This design achieves competitive results across multiple VLN benchmarks while notably lowering computational overhead.

**Dynam3D (Wang et al., 2025)** tackles the key shortcomings of applying Video-VLMs to real-world navigation, such as weak 3D reasoning, limited long-term memory, and poor adaptability to dynamic settings. It projects CLIP features into 3D space and builds hierarchical patch-, instance-, and zone-level representations that update online, enabling both geometric understanding and robust memory across changing environments. With large-scale 3D-language pretraining, Dynam3D achieves state-of-the-art results on multiple VLN benchmarks and shows strong potential for real-world deployment.

### A.1.4 ENVNAV BASELINES: 3D-MLLMs

**ChatScene (Huang et al., 2024)** reformulates 3D scene understanding by shifting focus from global scene embeddings to object-centric representations. It breaks scenes into object proposals with unique identifiers, allowing precise grounding and flexible interaction across tasks. This design unifies diverse 3D scene-language problems under a QA framework, yielding strong gains on multiple benchmarks with minimal fine-tuning.

**Reason3D (Huang et al., 2025b)** extends multimodal LLMs into richer 3D scene understanding by coupling language reasoning with dense visual outputs. Instead of stopping at text or numbers, it links point clouds and prompts to generate both responses and segmentation masks, supporting tasks like reasoning-driven segmentation, referring, and QA. A hierarchical mask decoder refines object predictions from coarse to fine, enabling more accurate comprehension of large, complex 3D scenes.

**LSceneLLM (Zhi et al., 2025)** tackles the challenge of extracting task-relevant details from dense 3D scenes by adaptively focusing on the most important regions. It uses an LLM-guided token selector to identify where to look, then applies a scene magnifier module to refine fine-grained details, combining them with global context for richer understanding. Alongside this framework, the authors introduce XR-Scene, a benchmark for cross-room scene understanding, where LSceneLLM achieves clear improvements over existing 3D-VLMs.

**3D-LLaVA (Deng et al., 2025)** is designed as a lightweight yet powerful assistant for 3D scene understanding and interaction. Instead of relying on multi-stage pipelines, it directly operates on point clouds through its Omni Superpoint Transformer, which unifies feature selection, visual prompt encoding, and mask generation. With hybrid pretraining and unified instruction tuning, 3D-LLaVA achieves strong results across multiple benchmarks while keeping the architecture simple and versatile.

**LLaVA-3D (Zhu et al., 2025)** extends the strong 2D priors of LLaVA into 3D scene understanding through a streamlined framework. By enriching 2D CLIP patches with 3D position embeddings, it forms 3D-aware patches that support accurate spatial outputs such as 3D bounding boxes. Joint

2D–3D instruction tuning enables a unified model that trains more efficiently than prior 3D LMMs, achieves state-of-the-art results on 3D tasks, and preserves robust 2D vision-language capabilities.

## A.2 IMPLEMENTATION DETAILS FOR BASELINES

We evaluate heterogeneous model families under a unified EnvNav task formulation and consistent metrics, including representation-based methods, VLN agents, and 3D-MLLMs. All models are tested using their officially released checkpoints and native pipelines, without any architectural modification or task-specific fine-tuning, in order to faithfully reflect their inherent capacity for global spatial reasoning in building-scale environments.

**Representation-based methods.** These methods are evaluated following their original object-goal navigation pipelines, which map language descriptions to 3D coordinates via learned 3D language fields. To preserve their intended functionality while enabling fair comparison under ERNav, we use an LLM only to extract the target landmark phrase from each instruction, which serves as the object-goal query required by these methods. The subsequent localization process strictly follows their original implementation, without any change to the model weights or architecture. For HOV-SG, which additionally supports structured instruction parsing, we report results for both its standard and navigation variants, following the official codebase.

**3D-MLLMs.** All 3D-MLLMs are evaluated in a zero-shot manner using their released inference pipelines, without further fine-tuning. This choice is made for two reasons: (i) most existing 3D-MLLMs do not provide practical training or fine-tuning utilities, and (ii) ERNav is designed to evaluate the intrinsic generalization and spatial reasoning ability of existing models rather than task-specific optimization. Although the proposed 3D-LangNav model is fine-tuned, its training data are independently generated by us and have no overlap with the ERNav scenes or instructions, ensuring that the comparison remains fair.

Despite the heterogeneity of model architectures, all methods are evaluated under identical conditions in EnvNav, requiring them to predict a 3D target location based on a natural language instruction. We rely on each method's native output format and reasoning pathway, and standardize only the evaluation metrics (e.g., NE, SR) for comparison. This protocol ensures both fairness and interpretability of the results in Tab. 3, while allowing each method family to operate in its most representative and intended manner.

## A.3 MORE EXPERIMENTS ON ENVEXP

We report additional results of different methods on EnvExp under varying thresholds $\theta$ in ObjCov, as shown in Tab. 6. We also provide statistics on steps, the efficiency term $E$, and a brute-force baseline that exhaustively visits all possible positions.

The results show that our dual-sighted exploration strategy introduces extra steps due to observing objects from multiple viewpoints, but this overhead is modest and acceptable, especially compared with the brute-force baseline. Moreover, even with $d_{far} = 3$ m fixed, DSS consistently outperforms other baselines across different $\theta$, demonstrating robustness to objects and scenes of varying sizes.

Table 6: Exploration metrics of different methods in EnvExp.

| Methods | Steps | Coverage%↑ | $M(3)\%↑$ | $M(5)\%↑$ | $M(10)\%↑$ | $E\%↑$ | ObjCov%↑ |
|---|---|---|---|---|---|---|---|
| FBE | 638 | 93.9 | 33.1 | 29.8 | 24.3 | **82.1** | 27.2 |
| ANS | 727 | 91.5 | 35.8 | 31.5 | 26.2 | 80.9 | 29.0 |
| PONI | 767 | 88.0 | 33.6 | 31.1 | 26.0 | 80.4 | 27.0 |
| LFE | 761 | 92.2 | 34.7 | 31.6 | 26.2 | 80.5 | 27.9 |
| Human | 1,279 | 96.2 | 31.1 | 26.1 | 20.3 | 74.7 | 23.2 |
| Brute-force | 19,935 | 100.0 | 100.0 | 100.0 | 100.0 | 0.0 | 0.0 |
| ANS+DSS | 2,104 | **99.4** | **60.7** | **54.7** | **47.0** | 67.5 | **41.0** |

Table 7: Influence of camera HFOV and height on ERNav performance.

| Setting | EnvExp (Cov) | EnvExp (ObjCov) | EnvExp (F1) | EnvRep (R@20) | EnvRep (MR) | EnvNav (SR) |
|---|---|---|---|---|---|---|
| Default (90°, 1.25m) | 99.4 | 41.0 | 52.9 | 84.2 | 12.1 | 50 |
| HFOV = 60° | 99.4 | 39.6 | 52.1 | 82.5 | 12.8 | 48 |
| HFOV = 120° | 99.4 | 38.2 | 53.3 | 83.8 | 12.3 | 49 |
| Height = 1.00m | 99.5 | 40.8 | 23.1 | 71.9 | 16.1 | 37 |
| Height = 1.50m | 99.2 | 40.5 | 54.4 | 84.0 | 12.2 | 50 |

Table 8: Influence of exploration quality on representation and navigation performance.

| Exploration Strategy | EnvExp (ObjCov) | EnvRep (R@20) | EnvRep (MR) | EnvNav (SR) |
|---|---|---|---|---|
| FBE | 27.2 | 72.6 | 18.4 | 39 |
| ANS | 29.0 | 78.3 | 15.2 | 44 |
| PONI | 27.0 | 71.5 | 21.9 | 37 |
| LFE | 27.9 | 77.1 | 16.2 | 45 |
| **Ours** | **41.0** | **84.2** | **12.1** | **50** |

## A.4 GENERALIZATION TO CAMERA CONFIGURATIONS

In ERNav, only the exploration stage directly depends on the camera configuration, while the representation and navigation stages operate purely on the reconstructed 3D map and the extracted features. To evaluate the robustness of our pipeline to different sensor configurations, we conduct an additional sensitivity analysis by varying the exploration camera parameters along two axes: horizontal field of view (HFOV) $\{60°, 90°, 120°\}$ and camera height $\{1.00m, 1.25m, 1.50m\}$, while keeping all other components fixed.

The results in Tab. 7 demonstrate that ERNav is highly robust to variations in camera configurations, with downstream navigation performance remaining within a narrow range. Varying the HFOV from 60° to 120° leads to only minor differences in Success Rate, indicating that the multi-view aggregation and Dual-Sighted Strategy (DSS) effectively mitigate changes in single-frame coverage and pixel density.

In contrast, setting the camera height to 1.00m results in a noticeable performance drop, especially in geometric F1 and navigation success. This is expected, as a lower viewpoint is more prone to occlusion by furniture, particularly for objects located on tables and higher surfaces, leading to reduced observability. Once the camera clears major occlusions (1.25m and 1.50m), the performance recovers and stabilizes, confirming that the method remains robust as long as sufficient visibility is preserved.

These results highlight an important structural advantage of ERNav: the decoupling between hardware-dependent exploration and representation-based navigation. Unlike end-to-end VLN agents that are often tightly coupled to specific sensor configurations, ERNav allows one platform to perform exploration and a different platform to navigate using the constructed map, making the system inherently robust to robot heterogeneity.

## A.5 INFLUENCE OF EARLY STAGES ON END-TO-END NAVIGATION

Since ERNav follows a modular pipeline, it is important to understand how the quality of earlier stages affects performance in downstream navigation. From the main results in Tabs. 1 to 3, we already observe a consistent dependency across stages. Specifically, higher exploration quality (Table 1) leads to improved 3D representations, while stronger representations (Table 2) consistently translate into better navigation performance (Table 3).

To explicitly quantify this dependency, we conduct a controlled experiment in which the representation and navigation modules are fixed (using the proposed 3D-LangNav architecture), while only the exploration strategy is varied. This allows us to directly measure how exploration quality propagates to the final task.

As shown in Tab. 8, there is a clear and monotonic relationship across stages. Higher object coverage in the exploration stage leads to more informative and complete scene representations, as reflected by higher Recall@20 and lower Mean Rank. In turn, these stronger representations directly improve navigation success. Low-coverage strategies are consistently bottlenecked at SR $\leq 39\%$, whereas our high-coverage strategy achieves a SR of $50\%$.

These results demonstrate that exploration quality directly determines the upper bound of representation quality, which in turn constrains navigation performance. Thus, the relatively lower navigation performance of baseline methods is primarily due to insufficient perceptual coverage in the exploration stage, rather than deficiencies in language reasoning.

### A.6 COMPLEXITY ANALYSIS OF 3D-LANGNAV.

We provide the complexity analysis to prove the advantage of our node-first, edge-verification strategy in 3D-LangNav. Let $n = |\mathcal{V}|$ be the node count in the scene graph $\mathcal{G}$, and $m$ the number of landmarks mentioned in the instruction graph $\mathcal{G}_{\text{inst}}$. Naive global matching approach is equivalent to the problem of subgraph isomorphism, which in the worst case requires checking all $\binom{n}{m} \cdot m!$ possible mappings, resulting in *exponential* complexity $\mathcal{O}(n^m \cdot m!)$, making it infeasible for large-scale environments in ERNav. Our divide-and-conquer strategy decomposes the matching into two stages. Each landmark is independently matched to a set of $k$ candidate nodes using vision-language similarity queries in the 3D language field, reducing the search space from $n^m$ combinations to $k^m$. Verifying spatial relationships for each candidate set requires $\mathcal{O}(m^2)$ pairwise checks, giving $\mathcal{O}(k^m \cdot m^2)$. Since $k \ll n$ in practice, the total complexity becomes $\mathcal{O}(m \cdot n \cdot c_q + k^m \cdot m^2 \cdot c_v)$, representing an exponential reduction in the search space compared to direct subgraph matching.

### A.7 REAL-WORLD APPLICATIONS

Once the target coordinates are predicted by ERNav, different strategies can be employed to guide the agent to the destination. We outline several complementary approaches below: **(1) Classical path planning.** Since exploration provides an occupancy map, the agent can directly plan a path from the start position to the target using standard shortest-path algorithms such as A* or Dijkstra. This approach leverages the geometric structure of the explored environment and produces efficient paths when the map is sufficiently complete. **(2) Graph-based navigation.** To maintain consistency with the discrete VLN setting, we can build a navigation graph over all navigable locations observed during exploration. The agent then traverses between viewpoints along graph edges, using the off-the-shelf point-goal navigation controller DD-PPO (Wijmans et al., 2019) to execute local motions between consecutive nodes. **(3) Trajectory replay.** Exploration inherently yields a complete walking trajectory around the building, staying within 1 meter of the navigation boundary. Therefore, a simple but effective strategy is to first move the agent to the closest point on this trajectory, and then follow it until reaching the location closest to the predicted target. This avoids redundant planning and guarantees connectivity to most navigable regions. **(4) Cross-level and failure recovery.** For vertical transitions (e.g., stairs, elevators), the agent reuses the paths observed during exploration, ensuring reliable cross-level navigation. In cases where the agent becomes stuck near obstacles, we employ a lightweight heuristic: the agent rotates in place and attempts to move forward, after which control is handed back to DD-PPO for local correction. Together, these strategies provide a flexible toolkit for integrating ERNav with both classical and learned navigation pipelines, enabling robust execution across different environment configurations.

### A.8 EXPLORATION TRAJECTORY VISUALIZATIONS

We further visualize the exploration trajectories generated by different methods in Fig. 3. With the integration of our DSS, the agent captures both close-up views of objects for detailed information and distant views for relational context, thereby achieving the highest map coverage as well as the highest ObjCov.

### A.9 PROMPT TEMPLATE FOR 3D-LANGNAV

In this section, we provide the prompt templates used in our 3D-LangNav, with the system prompt shown in Fig. 4 and the user prompt in Fig. 5.

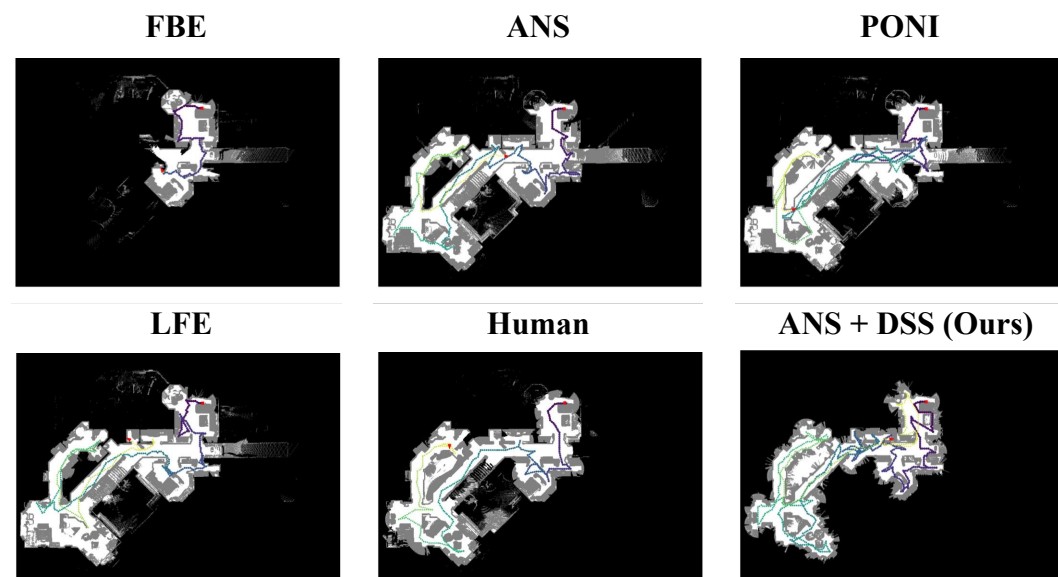

Figure 3: Visualization of exploration trajectories generated by different methods.

---

**System Prompt for 3D-LangNav**

You are an advanced 3D environment understanding assistant. Your main objective is to interpret a language-based instruction describing an indoor environment and identify which candidate landmark best matches the specified target.

The inputs are presented in the following formats:

1. **Instruction**: A natural language description involving spatial relationships (e.g., relative positions, distances) among landmarks.
2. **Candidates**: A list of candidates for all landmarks mentioned in the instruction. Each landmark belongs to one of the following types:
   - Floor: No explicit candidate data is provided. The positions belonging to this floor must be inferred from the connectivity map. (Note: floor index starts from 1.)
   - Room: The candidates are a list of nodes in the connectivity map.
   - Object: Includes the object's unique identifier and the 3D coordinates of its center.
3. **Target**: The specific landmark name within the instruction that must be located among the given candidates.
4. **Navigable Nodes**: A representation of the environment layout, including a list of key positions.
5. **Start Position**: The agent's initial 3D location, which may be referenced in the instruction.

**Coordinate System**

All 3D positions (x, y, z) follow the convention:
   - x-axis: Left to right, increasing to the right.
   - y-axis: Floor to ceiling, increasing upward.
   - z-axis: Front to back, increasing forward.

**Your Task**

Analyze the provided information to decide which candidate is the correct match for the target. Consider all clues from the natural language description, especially spatial relationships, and compare them with the bounding boxes and 3D positions of the candidates.

**Output Format**

You must identify a single candidate as the correct match in the following format:
   "The correct candidate is <Candidate_ID>."

Figure 4: System Prompt for spatial reasoning in 3D-LangNav.

## A.10    DYNAMIC SCENES

The current version of ERNav focuses on static indoor environments, and we explicitly acknowledge this as a limitation of the benchmark. At the same time, we emphasize that the absence of dynamic scenes does not compromise the effectiveness or validity of ERNav for evaluating embodied navigation systems. ERNav is designed to assess an agent's ability to explore environments, construct spatial representations, and reason over them to perform navigation, which remains a fundamental capability even when the environment undergoes future changes.

In practice, dynamic scenes can be viewed as updated versions of previously observed environments, where the core challenges of exploration, representation, and navigation remain unchanged. This

---

**User Prompt for 3D-LangNav**

1. **Instruction**: {instruction}
2. **Candidates**:
   - Landmark 0: {landmark_name} ({landmark_type})
     - Candidate 0: {pos: {candidate_position}, bounding box: {candidate_bounding_box}}
     - Candidate 1: {pos: {candidate_position}, bounding box: {candidate_bounding_box}}
     ...
   - Landmark 1: {landmark_name} ({landmark_type})
     ...
3. **Target**: {target_name}
4. **Navigable Nodes**:
   - Node 0: {node position}
   - Node 1: {node position}
   ...
5. **Start Position**: {start_pos}

---

Figure 5: User prompt template for spatial reasoning in 3D-LangNav.

assumption is consistent with prior work in embodied navigation; for example, ObVLN (Hong et al., 2024) points out that most existing VLN benchmarks also assume static environments, yet they continue to serve as effective testbeds for evaluating navigation capabilities. Similarly, the broader map-and-plan literature has shown that dynamic changes are often handled through incremental updates rather than requiring a fundamentally different benchmark design. Existing works have explored mechanisms such as continuous map refinement, change detection, and selective re-exploration to handle environmental dynamics (Huang et al., 2025a; Gu et al., 2024).

ERNav naturally supports such extensions. An agent can update its internal representation during EnvNav or trigger partial re-exploration when observed discrepancies exceed a predefined threshold. These directions represent important future work that will further enhance the real-world applicability of the benchmark. Importantly, they are orthogonal to the core contribution of ERNav, which lies in providing the first unified benchmark that explicitly evaluates how exploration and representation jointly influence downstream navigation performance at building scale.

## A.11 NOISES IN ERNAV

The noises in ERNav mainly comes from the inherent imperfections and artifacts present in real-world RGB-D scans (e.g., Matterport3D (Chang et al., 2017)), rather than synthetic noise artificially injected into clean data. This distinction is well recognized in the 3D scene understanding literature (Dai et al., 2017), where real scanning data is characterized by natural noise patterns and occlusions that differ fundamentally from synthetic environments.

In ERNav, these realistic noise sources mainly arise from two aspects. First, commodity RGB-D sensors introduce depth quantization errors, boundary noise, and missing measurements, particularly on reflective or transparent surfaces. Second, since agents construct representations from partial, egocentric observations during exploration (instead of relying on pre-processed global point clouds), the reconstructed data naturally contains artifacts such as outliers, floaters, and occlusion-induced incompleteness. In contrast, camera poses are not treated as a source of noise in our benchmark, as we follow Habitat's standard protocol and directly use the simulator-provided sensor poses.

Although we do not perform a synthetic noise ablation study, ERNav is explicitly designed to be robust to these realistic imperfections. This robustness is achieved through three key mechanisms: (i) multi-view feature aggregation to reduce transient sensor noise, (ii) density-based outlier rejection (DBSCAN) to filter geometric artifacts, and (iii) hierarchical abstraction into object- and region-level representations, which smooths high-frequency noise before reasoning. Our experimental results in

Tab. 2 and Tab. 3 demonstrate that the proposed method can reliably reconstruct and reason over such imperfect observations.

## A.12 LLM Usage Statement

LLMs were used in this work only as a writing-assist tool. Their role was limited to checking grammar, polishing language, and verifying formatting consistency. They were **not** used for research ideation, content generation, data analysis, or development of results. All ideas, methodologies, and conclusions presented in this paper were conceived and written by the authors. The authors take full responsibility for the contents of the manuscript.

## A.13 Discussion

**Limitations.** Although 3D-LangNav achieves competitive performance, several limitations remain. First, the current framework cannot handle dynamic environments, where objects or layouts may change after exploration, which can cause failures in target localization. Second, the representation construction process is not real-time, limiting applicability to time-sensitive robotic tasks. Moreover, deploying a 72B model on real robots is impractical under current hardware constraints. Finally, because the method relies on candidate selection for prediction, it may still miss valid targets in cluttered or ambiguous scenes, resulting in performance degradation.

**Future Work.** In future work, we aim to address these limitations in several directions. We plan to leverage more advanced LLMs with full finetuning to further enhance instruction grounding and coordinate prediction. We will also expand the set of baselines for ERNav by incorporating NeRF- and 3DGS-based scene representations as well as video-based navigation methods, providing a more comprehensive evaluation. Moreover, we intend to fine-tune a multimodal LLM under our current candidate-filtering framework, improving its ability to identify and retain true target objects more reliably.

