# OpenReview forum: "ERNav: A Unified, Realistic Benchmark for Embodied AI with Exploration, Representation, and Navigation"
_ICLR.cc/2026/Conference — Submitted to ICLR 2026_

### Official Review · Reviewer_oc3P · 2025-10-25

**Soundness:** 3
**Presentation:** 2
**Contribution:** 3
**Rating:** 4
**Confidence:** 3

**Summary:**

This paper introduces ERNav, a unified benchmark designed to integrate Exploration, Representation, and Navigation into a single embodied AI evaluation framework. The benchmark defines three sub-tasks—EnvExp, EnvRep, and EnvNav—to separately assess exploration ability, representation quality, and navigation performance, respectively. The authors also propose a baseline model, 3D-LangNav, which jointly leverages language and 3D perception for embodied navigation. Extensive experiments across multiple embodied AI models and 3D-LLMs are reported, aiming to establish a standardized evaluation protocol for realistic embodied reasoning and navigation.

**Strengths:**

1. Timely and relevant topic.
The work tackles an important and emerging direction in embodied AI—unifying exploration, reconstruction, and navigation—which are often studied in isolation.

2. Clear task formulation.
The benchmark design into three sub-tasks (EnvExp, EnvRep, EnvNav) is logical and well-motivated. The inclusion of EnvRep, in particular, provides a valuable bridge between perception and action evaluation.

3. Informative baseline.
The proposed 3D-LangNav baseline achieves reasonable results across multiple tasks and offers insights into the interplay between language grounding and navigation performance.

**Weaknesses:**

1. Limited related work discussion.
The paper omits discussion of closely related efforts such as MSR3D [1], which already provides a comprehensive evaluation protocol connecting scene understanding and navigation. The relationship and differences between ERNav and such benchmarks need clarification.

2. Lack of dataset statistics and quality assessment.
As a benchmark paper, it is important to include dataset composition, diversity metrics, and quality control analysis. Without these, it is difficult to judge the representativeness or reliability of the proposed benchmark.

3. Unclear experimental settings.
The description of how representation-based methods and 3D-LLMs are adapted for navigation tasks is vague. The implementation details, training configurations, and evaluation criteria should be explained more explicitly.

4. Ambiguous comparisons in Table 3.
The results in Table 3 mix heterogeneous model categories (representation-based, language-based, 3D-LLMs) without clarifying the evaluation setup. This makes it hard to interpret relative performance. Improving table organization and adding clearer experimental notes would enhance readability and fairness.

[1] Multi-modal Situated Reasoning in 3D Scenes

**Questions:**

1. How are the representation-based methods in Table 3 integrated into the navigation framework? Are they directly used as frozen feature extractors, or fine-tuned for embodied tasks?

2. Were the 3D-LLMs in Table 3 fine-tuned on the same navigation trajectories or pretrained independently?

3. How large is the dataset used for each sub-task (EnvExp, EnvRep, EnvNav), and how is data quality validated?

4. Could the authors elaborate on the differences between ERNav and MSR3D, particularly in evaluation design and practical difficulty?

---

> ### Author Response · Authors · 2025-11-22
> **Resposne to Reviewer oc3P - Part 1**
>
> We thank Reviewer oc3P for taking the time to review our paper and for the constructive feedback provided. Please refer to the following responses to address your comments.
>
> ---
> > ### Weakness 1. Related Work
>
> We thank the reviewer for highlighting this omission and for pointing us to MSR3D. MSR3D is indeed an influential and valuable benchmark that connects scene understanding with navigation, and we appreciate the opportunity to better situate our work relative to it. While ERNav and MSR3D share the high-level goal of evaluating embodied scene understanding, they differ in several fundamental aspects, including task scale, required representations, and evaluation design. Since Question 4 also explicitly asks about the differences between ERNav and MSR3D, we provide a detailed, point-by-point comparison in our response to that question to avoid redundancy.
>
> We will ensure that the revised paper includes a clear discussion of MSR3D in the related work section, explicitly contrasting it with ERNav and clarifying our respective scopes and contributions.
>
> > ### Weakness 2. Dataset Statistics
>
> We thank the reviewer for raising this important point. Since ERNav reorganizes and integrates data from well-established benchmarks rather than collecting new raw data, we did not originally highlight dataset composition and quality assessment as standalone contributions. However, we agree that providing explicit dataset statistics and quality descriptions will significantly improve the clarity and completeness of a benchmark paper. As explained in Question 3, the three ERNav subtasks are constructed from high-quality, widely used datasets。 EnvExp is derived from the Matterport3D validation and test splits, covering 29 scenes (61 floors), with 183 exploration episodes. Data quality fully inherits the reliability of the Matterport3D dataset. EnvRep and EnvNav rely on the REVERIE unseen validation split, consisting of 10 large indoor scenes and 3,573 instruction–trajectory pairs, complemented with region-level landmarks from the training split. Because all components are sourced from rigorously validated datasets (Matterport3D and REVERIE), the reliability and representativeness of the data are well established.
> Following the reviewer’s suggestion, we will add a dedicated subsection summarizing dataset statistics, diversity characteristics, and data provenance, and include a table that clearly lists the composition of each sub-task. We appreciate the reviewer for pointing out this omission and will ensure it is fully addressed in the revised version.
>
> > ### Weakness 3. Experimental Settings
>
> We thank the reviewer for pointing out the need for clearer experimental settings. Since both the representation-based methods and 3D-LLMs are evaluated directly using their released checkpoints and native pipelines with no additional fine-tuning or architectural modifications, we initially kept the description brief. We agree that further clarification will enhance transparency. In response to **Question 1 and Question 2**, we provide detailed explanations of how these models are adapted for ERNav. In the revised version, we will include a dedicated subsection to present these implementation details and evaluation criteria to ensure full clarity and reproducibility.
>
> > ### Weakness 4. Comparison in Table 3.
>
> We appreciate the reviewer’s feedback regarding Table 3. This concern is closely related to Question 1 (representation-based methods) and Question 2 (3D-LLMs), where we provide detailed explanations of how each model family is evaluated under ERNav. Although Table 3 contains heterogeneous model categories, the comparison is not ambiguous: all models are evaluated under the same EnvNav task formulation and following the same evaluation metrics. Since we rely on each method’s native pipeline without additional fine-tuning, the results reflect their inherent capacity for global spatial reasoning in building-scale environments.
>
> We agree that the presentation can be improved. In the revised version, we will restructure Table 3 and add clearer experimental notes so that the evaluation setup and the relationship between different model families are easier to interpret.

---

> ### Author Response · Authors · 2025-11-22
> **Resposne to Reviewer oc3P - Part 2**
>
> > ### Question 1. Representation-based methods in Table 3
>
> We appreciate the reviewer’s question. The representation-based methods in Table 3 are not used as a frozen feature extractor or fine-tuned for embodied navigation. Instead, we follow their original object-goal navigation pipelines. These methods construct 3D language fields that directly map language descriptions to 3D coordinates, which is exactly how they support object-goal navigation in their original papers. However, they are not designed to interpret the complex relational instructions required in EnvNav (e.g., multi-object relations, global spatial constraints). To ensure fair evaluation while preserving their intended functionality, we use an LLM only to extract the target landmark phrase from the full instruction. This keeps the representation intact and simply provides these methods with the object-goal query they require, after which we apply their original localization pipeline without any fine-tuning or architectural changes. For HOV-SG, which provides an additional heuristic module for handling complex instructions, we report results for both variants following the official implementation (shown as the “Nav” rows in Table 3).
>
> > ### Question 2. 3D-LLMs in Table 3
>
> We did not modify or fine-tune the 3D-LLMs reported in Table 3 and directly evaluate their released models on ERNav for two reasons. First, many 3D-LLMs only provide inference code without any training or fine-tuning pipelines, making fine-tuning practically infeasible. In addition, fine-tuning such large models is computationally expensive and requires extensive hyperparameter search, which is beyond the scope of this benchmark. Second, ERNav is intended to assess the intrinsic ability of existing 3D-LLMs to perform global spatial reasoning and understand complex navigation instructions without task-specific retraining. Evaluating their original models aligns with the benchmark’s goal.
> We also note that while 3D-LangNav is fine-tuned, its training scenes and instructions do not overlap with any data in ERNav. The training data are generated by us and form part of our contribution. Thus, the comparison remains fair. We will clarify this more explicitly in the revised paper.
>
> > ### Question 3.
>
> We thank the reviewer for the question. Since each sub-task in ERNav is constructed by reorganizing well-established datasets, the quality of the underlying data is inherently guaranteed, and we therefore did not emphasize “new data collection” as a contribution of the benchmark.
>
> 1. EnvExp.
> As described in lines 183 and 360 of the paper, EnvExp is built upon the validation and test scenes of Matterport3D. We perform floor-level exploration over 29 scenes (61 floors). For each floor, we randomly sample three starting positions, producing 183 exploration episodes in total. Since all scenes come directly from Matterport3D, the data quality is fully inherited from this widely used standard dataset.
>
> 2. EnvRep and EnvNav.
> Both tasks use the REVERIE validation unseen split, which contains 10 unseen scenes and 3,573 trajectory–instruction pairs, which is directly used in EnvNav. As noted in line 363 of the paper, the unseen scenes contain relatively few region-level landmarks. To ensure sufficient coverage, we additionally incorporate the region-level landmarks from the REVERIE training split, resulting in 3,297 object-level landmarks and 1,324 region-level landmarks used in EnvRep. All data come directly from REVERIE’s official annotations, which have been widely adopted and validated in prior VLN works.
>
> To address the reviewer’s concern, we will add a table summarizing dataset statistics and clarify data sources and quality validation in the revised version.

---

> ### Author Response · Authors · 2025-11-22
> **Resposne to Reviewer oc3P - Part 3**
>
> > ### Question 4. Differences between ERNav and MSR3D
>
> MSR3D is indeed a strong and influential benchmark that significantly advances 3D scene understanding by providing multi-modal inputs, diverse scenes, and a large number of QA-style tasks covering recognition, spatial reasoning, and short-range navigation. We acknowledge its value, and in the revised version we will add a dedicated discussion comparing MSR3D with ERNav.
>
> However, despite the conceptual connections, ERNav and MSR3D differ fundamentally in evaluation design, task emphasis, and practical difficulty, as summarized below.
>
> **1. Scene scale and difficulty emphasis**
> MSR3D focuses on scene diversity, drawing from ScanNet, 3RScan, and ARKitScenes. These scenes typically correspond to single rooms or a few connected rooms, which provide excellent coverage but remain local-scale (especially, all navigation scenes are from ScanNet). In contrast, ERNav emphasizes scene scale: all tasks are conducted in building-level Matterport3D environments, each containing multiple floors, dozens of rooms, and long-range connectivity. The spatial scale in ERNav is roughly an order of magnitude larger, making global reasoning substantially more challenging.
>
> **2. Instruction characteristics and reasoning scope**
> Due to its scene scale, MSR3D naturally focuses on short-range navigation queries and a broad set of multimodal tasks (object descriptions, counting, existence, etc.). The navigation-related items typically involve within-room, short-distance behaviors. ERNav, designed specifically for embodied navigation, uses long-horizon, cross-room natural-language instructions, with larger spatial extent and denser spatial relations. The instructions often require global spatial reasoning, understanding long-range connectivity, and navigating across multiple floors—capabilities that MSR3D does not target.
>
> **3. Action space and task formulation**
> MSR3D operates in a single-step, situated-navigation style: the model predicts step-wise actions such as "turn right", similar to classic VLN controls. ERNav adopts a different paradigm. After active exploration, the model must perform global path reasoning and coordinate prediction. Its action space is continuous 3D target coordinates, shifting the difficulty from step-level control to global scene understanding, multi-modal alignment, and target localization. Thus, both benchmarks measure navigation ability, but from entirely different perspectives.
>
> **4. Input modality and pipeline realism**
> MSR3D, like many scene-understanding benchmarks, uses ground-truth point clouds as input. These are clean, complete, and do not require exploration. ERNav explicitly includes autonomous exploration and mapping, which are absent in MSR3D. The agent must reconstruct the scene from RGB observations, producing imperfect, noisy point clouds, which are closer to real-world robotics pipelines. This design directly evaluates robustness to reconstruction noise and long-range spatial understanding.
>
> **5. Evaluation design and output format.**
> MSR3D evaluates navigation through single-step, low-level action predictions. As a result, the navigation component is cast as a multi-class classification problem, and performance is measured using accuracy over discrete action choices. This aligns with MSR3D’s emphasis on situated, step-wise action understanding rather than full navigation trajectories. In contrast, ERNav requires the model to predict a continuous 3D coordinate representing the final navigation target after global reasoning over a building-scale map. This formulation allows ERNav to adopt standard VLN navigation metrics, including Navigation Error (NE) and Success Rate (SR), which evaluate the agent’s ability to understand global spatial structure and reach the correct destination. Therefore, the evaluation design of ERNav emphasizes trajectory-level goal understanding and map-based reasoning, which is not captured by MSR3D’s step-level action classification framework.
>
> In summary, although both benchmarks involve 3D perception and navigation, their goals, assumptions, and difficulties are fundamentally different and complementary. We will include a detailed comparison with MSR3D in the revised paper to make these distinctions clearer.

---

> ### Author Response · Authors · 2025-11-28
> **Kind Request for Discussion and Reconsideration of the Score**
>
> We sincerely thank Reviewer oc3P for the valuable feedback and insightful suggestions that have helped us significantly refine our work.
>
> We have posted a detailed response and uploaded a revised manuscript, which includes new experiments and crucial clarifications addressing all the concerns you raised.
>
> As the discussion period is drawing to a close, we would be very grateful if you could confirm whether our responses and revisions have satisfactorily resolved your initial concerns. Your final input is invaluable for ensuring the quality and clarity of our work, and we stand ready to engage in any further discussion.

---

### Official Review · Reviewer_zBhR · 2025-10-30

**Soundness:** 3
**Presentation:** 2
**Contribution:** 3
**Rating:** 4
**Confidence:** 3

**Summary:**

This paper introduces ERNav, a new benchmark that incorporates exploration, map construction and navigation at building level. The proposed benchmark introduces a higher level of challenge as it better mimics the complex real-world scenario. The paper also introduces a strong baseline 3D-LangNav that can address the above mentioned 3 points simultaneously. Extensive experiments and comparison with baseline models suggested that ERNav poses challenge to existing methods.

**Strengths:**

1. The paper addresses the long-standing problem where embodied AI benchmarks and 3D vision benchmarks are not fully aligned with the real-world setting.

2. Extensive experiments show that there still exist gaps between the existing methods and the more realistic setting.

3. Building level benchmarks could inspire research towards more realistic settings.

**Weaknesses:**

1. The third subtask provided by the benchmark, i.e. EnvNav, seems to be too easy if standalone. The model is only asked to provide a coordinate, without actual execution in the scene, making the input coordinates useless since without it the model can also predict a target coordinate.

2. Therefore, it is questionable whether the provided benchmark is more challenging since all the actual navigation done in VLN works are simplified when given actual instructions.

3. The benchmark doesn't take into consideration the dynamic scenes, where objects locations can vary largely, making the proposed task setting less efficient.

**Questions:**

See Weaknesses.

---

> ### Author Response · Authors · 2025-11-22
> **Response to Reviewer zBhR - Part 1**
>
> We appreciate Reviewer zBhR for the time and effort in reviewing our paper and offering constructive feedback. Please find our responses to the comments below.
>
> ---
> > ### Weakness 1. The difficulty of EnvNav
>
> We respectfully clarify that the EnvNav task is not easy when evaluated standalone, nor are the input coordinates redundant. The design of EnvNav is centered on evaluating high-level, global 3D reasoning, which is the core scientific challenge.
>
> **(1) EnvNav is inherently challenging and non-trivial.**
> The experimental results in Table 3 directly refute the notion that EnvNav is easy. All current baselines, particularly 3D-MLLMs designed for spatial reasoning, perform poorly, with none achieving a Success Rate (SR) above 20%. EnvNav requires inferring a target location across an entire building-scale environment using instructions containing complex spatial relations. This demands global 3D reasoning—a capability recent studies consistently show current MLLMs struggle with [1,2].
>
> **(2) Predicting global coordinates is substantially harder than traditional VLN actions.**
> Standard VLN involves local step-wise actions (“forward/turn”), grounded in continuous first-person perception. In contrast, EnvNav requires direct inference of a global target coordinate over a much larger spatial space. The suggestion that the model could predict coordinates without coordinate grounding is analogous to assuming a VLN agent could navigate without observations, which is inconsistent with embodied navigation. We do not include physical execution because point-goal navigation is already solvable once the occupancy map is available; shortest-path planning introduces no additional conceptual difficulty (as discussed in Section A.4), and the scientific challenge of EnvNav lies in reasoning, not low-level control.
>
> **(3) EnvNav is substantially more complex than existing 3D scene-language benchmarks.**
> ScanRefer[3], a widely used 3D grounding benchmark, operates on small indoor scenes, provides ground-truth instance masks and IDs, and frames the task as simple object ID selection. EnvNav instead spans full buildings, contains richer and more complex spatial instructions, and offers no GT object metadata. Both the spatial scale and reasoning complexity are significantly higher, making EnvNav a meaningfully harder challenge.
>
> **(4) Coordinate grounding is essential, not redundant.**
> The region/object coordinates predicted in EnvRep provide the spatial interface required for downstream reasoning. Removing this grounding would eliminate the geometric structure that the model must rely on. Evidence comes from ChatScene [4], which predicts coordinates without map-based grounding and achieves only an 8% SR—highlighting that coordinate grounding is crucial rather than optional.
>
> [1] Chen, Boyuan, et al. "Spatialvlm: Endowing vision-language models with spatial reasoning capabilities." CVPR. 2024.
>
> [2] Ma, Chenyang, et al. "Spatialpin: Enhancing spatial reasoning capabilities of vision-language models through prompting and interacting 3d priors." NeurIPS. 2024.
>
> [3] Chen, Dave Zhenyu, Angel X. Chang, and Matthias Nießner. "Scanrefer: 3d object localization in rgb-d scans using natural language." ECCV. 2020.
>
> [4] Huang, Haifeng, et al. "Chat-scene: Bridging 3d scene and large language models with object identifiers." NeurIPS. 2024.
>
> > ### Weakness 2. Challenges of ERNav
>
> We respectfully clarify that ERNav is more challenging than traditional VLN in several fundamental ways, both by introducing new required capabilities and by increasing the cognitive complexity of the final navigation task.
>
> **(1) ERNav requires capabilities that VLN agents do not support.**
> ERNav consists of three tightly coupled stages—exploration, representation, and navigation—each involving distinct and non-trivial challenges:
>     - Exploration: requires comprehensive, efficient environment coverage and balanced object observations, which we evaluate with the novel ObjCov metric.
>     - Representation: requires constructing consistent object-level and region-level semantic maps and performing 3D segmentation and scene abstraction.
>     - Navigation: requires global reasoning over building-scale environments and aligning instructions with a hierarchical 3D map.
> None of these capabilities is required—or supported—by standard VLN agents, which operate under a fundamentally different assumption (step-wise local perception + ground-truth access to continuous observations). Thus, ERNav introduces challenges that do not exist in conventional VLN.

---

> ### Author Response · Authors · 2025-11-22
> **Response to Reviewer zBhR - Part 2**
>
> **(2) Even when considering only the navigation stage, ERNav is strictly more difficult.**
> Traditional VLN focuses on local sequential decision making, whereas EnvNav requires global coordinate prediction from high-level relational instructions over an entire building-scale map. This demands multi-level spatial reasoning, long-range referencing, and large-scale multimodal alignment, as evidenced by the very low SR of 3D-MLLMs in Table 3. Therefore, EnvNav is not a simplified version of VLN navigation—it is a more abstract, more global, and more complex reasoning problem.
>
> **(3) The reviewer’s concern about “actual navigation simplification” does not apply in our paradigm.**
> As discussed in Weakness 1, once the agent has a complete occupancy map, point-goal execution introduces no conceptual difficulty. This is why map-and-plan systems typically rely on classical shortest-path planning after goal inference. Our benchmark focuses on the hard part—goal inference from environment knowledge—not on low-level control, which offers little research value in this setting.
>
> > ### Weakness 3. Dynamic Scenes
>
> We appreciate the reviewer’s observation. Our current benchmark focuses on static indoor environments, and we explicitly acknowledge this limitation in the paper (Line 1016), identifying dynamic scenes as an important direction for future extension.
>
> However, the absence of dynamic scenes does not reduce the efficiency or validity of our benchmark setting. ERNav is designed to evaluate an agent’s ability to explore, construct a representation, and navigate, which remains a fundamental capability even when object locations later change. Dynamic scenes can be viewed as an updated version of the static scene, and the core requirement remains unchanged. Similar precedents exist in embodied navigation research: for example, ObVLN[1] points out that current VLN benchmarks assume static environments and do not model instruction failures due to scene changes, yet these benchmarks remain widely adopted and effective for evaluating navigation agents.
>
> Moreover, in the broader map-and-plan literature, dynamic scenes are typically handled through incremental updates rather than requiring a fundamentally different benchmark design. Prior works[2-3] already provide mechanisms such as continuous map refinement, change detection, or selective re-exploration (e.g., ConceptGraphs[3] supports online graph updates when the environment differs from past observations). In ERNav, such mechanisms can be incorporated naturally: an agent may update its map during EnvNav or re-explore when discrepancies exceed a threshold. These extensions enhance real-world applicability, but they are orthogonal to our core contribution—providing the first benchmark that evaluates how exploration and representation jointly influence downstream navigation.
>
> In summary, dynamic scenes are an important future direction (and we plan to include them), but their absence does not compromise the validity, efficiency, or contribution of ERNav as a benchmark.
>
> [1] Hong, Haodong, et al. "Navigating beyond instructions: Vision-and-language navigation in obstructed environments." ACM MM. 2024.
>
> [2] Huang, Chenguang, et al. "BYE: Build Your Encoder with One Sequence of Exploration Data for Long-Term Dynamic Scene Understanding." IEEE Robotics and Automation Letters. 2025.
>
> [3] Gu, Qiao, et al. "Conceptgraphs: Open-vocabulary 3d scene graphs for perception and planning." ICRA. 2024.

---

> ### Author Response · Authors · 2025-11-28
> **Kind Request for Discussion and Reconsideration of the Score**
>
> We sincerely thank Reviewer zBhR for the valuable feedback and insightful suggestions that have helped us significantly refine our work.
>
> We have posted a detailed response and uploaded a revised manuscript, which includes new experiments and crucial clarifications addressing all the concerns you raised.
>
> As the discussion period is drawing to a close, we would be very grateful if you could confirm whether our responses and revisions have satisfactorily resolved your initial concerns. Your final input is invaluable for ensuring the quality and clarity of our work, and we stand ready to engage in any further discussion.

---

### Official Review · Reviewer_HHyV · 2025-11-01

**Soundness:** 3
**Presentation:** 4
**Contribution:** 3
**Rating:** 6
**Confidence:** 4

**Summary:**

This work proposes a new benchmark for vision-language navigation (VLN) called ERNav. The benchmark includes three subtasks: scene exploration, global scene representation construction after exploration, and target location identification. Separate metrics are designed for each subtask. In addition to the ERNav benchmark, this work also introduces a baseline model named LangNav. Experimental results show that ERNav presents new challenges for existing methods, while 3DLangNav achieves strong performance.

**Strengths:**

1. Existing embodied AI benchmarks indeed have many limitations and remain far from real-world applications. This work clearly identifies these limitations and proposes a new benchmark to address them. The motivation is well-grounded and appreciated, and the problem studied in this paper is both important and timely.
2. The writing is clear and easy to follow.
3. For the new benchmark, the authors conducted comprehensive experiments to evaluate existing methods.
4. The paper also introduces a baseline that outperforms many previous approaches.

**Weaknesses:**

1. Although the proposed benchmark includes three stages—exploration, representation, and navigation—the ultimate goal in real-world applications remains navigation. Beyond providing separate metrics for each subtask, it is more meaningful to investigate how the results of the first two subtasks influence the final navigation performance. This aspect is missing from the paper. The authors also claim that the benchmark is end-to-end, but in my view, it is actually modular, since each subtask is evaluated independently without a unified metric that supports end-to-end performance analysis.
2. It would be useful to support methods that can perform representation and navigation but not exploration (e.g. ConceptGraphs), by incorporating a rule-based exploration module. This would allow those methods to participate in all three subtasks and enable a fairer comparison in Table 3.
3. For a new benchmark in embodied AI, it would be valuable to measure human performance on the proposed tasks, providing the community with an estimate of the potential upper bound.
4. It would also be beneficial to include demonstrations and a discussion of current method limitations, along with potential future research directions.

**Questions:**

1. Lines 67–71 state that ERNav requires the model to explore the scene based on noisy RGB-D inputs, but I could not find any explanation in the paper describing how the benchmark introduces noise to the RGB-D inputs or ensures that the noise realistically reflects real-world conditions.
2. Line 212: It is unclear why the authors use L1-CD instead of L2-CD for evaluation.

---

> ### Author Response · Authors · 2025-11-22
> **Response to Reviewer HHyV - Part 1**
>
> We are grateful to Reviewer HHyV for dedicating time and effort to reviewing our paper and providing thoughtful and constructive feedback. Our detailed responses to the comments are outlined below.
>
> ---
> > ### Weakness 1. Influence of Early Stages & End-to-End Evaluation
>
> (1) **Influence of exploration and representation on Navigation.**
>
> We agree that quantifying how exploration and representation quality affect navigation is important. In fact, ERNav already partially reflects this dependency across three tables:
> - Table 1, Exploration quality (Cov & ObjCov) correlates with reconstruction performance. Methods achieving higher coverage consistently obtain better geometric reconstruction. This establishes the link between exploration and representation.
> - Table 2, Comparing prediction-based vs. GT-based segmentation reveals that representation quality is critical for both object-level and region-level retrieval, showing how perceptual noise affects downstream reasoning.
> - Table 3, Representation-based models with stronger scene construction produce better navigation accuracy, demonstrating how different map-building strategies translate into different levels of navigational performance.
>
> To make this dependency clearer, we conducted a controlled experiment where we fixed the Representation and Navigation modules (using our 3D-LangNav architecture) and varied the Exploration Strategy. This allows us to directly measure how the quality of the first stage propagates to the final stage. The results clearly demonstrate a strong monotonic correlation:
>
> **Table 1. Influence of early stages on final navigation performance.**
> | Exploration Strategy | EnvExp (ObjCov %) | EnvRep (R@20 %) | EnvRep (MR %) | EnvNav (SR %) |
> |----------------------|-------------------:|-----------------:|---------------:|---------------:|
> | FBE                  |               27.2 |             72.6 |           18.4 |             39 |
> | ANS                  |               29.0 |             78.3 |           15.2 |             44 |
> | PONI                 |               27.0 |             71.5 |           21.9 |             37 |
> | LFE                  |               27.9 |             77.1 |           16.2 |             45 |
> | **Ours**             |         **41.0** |       **84.2** |     **12.1** |       **50** |
>
> This table explicitly quantifies the stage dependency. From Exploration to Representation, there is a clear, monotonic trend where higher ObjCov leads directly to better representation quality, evidenced by higher Recall@20 and a significantly lower Mean Rank. From Representation to Navigation, this improved representation quality directly translates into final task success. The low-coverage methods (PONI, FBE) are bottlenecked at SR $\le 39\%$, while our high-coverage strategy achieves $50\%$ SR.This confirms that exploration quality directly governs the upper bound of EnvRep, which subsequently constrains EnvNav. The low performance of baselines on EnvNav is not due to poor reasoning, but due to a failure to gather sufficient perception data in the EnvExp stage.
>
> (2) **Clarification of “end-to-end” vs. “modular.”**
>
> We appreciate the reviewer’s perspective and understand the source of confusion. Our use of the term “end-to-end” does not refer to executing the entire benchmark in a single monolithic forward pass. Instead, it refers to the agent's ability to autonomously complete the full embodied navigation pipeline without external input or handcrafted components. This capability is absent in existing VLN benchmarks, which evaluate only the navigation stage and therefore cannot assess map-and-plan systems or study how perception errors propagate into navigation.
>
> ERNav intentionally supports both:
> - Modular evaluation, so that existing methods (e.g., VLN models without exploration, representation-based models without navigation) can still participate fairly.
> - Full-pipeline evaluation, for methods like 3D-LangNav that can perform all three stages.
>
> The decomposition into three subtasks serves two key purposes:
> 1. Fairness and inclusiveness: Most prior works do not cover all stages, and separating the tasks allows them to be evaluated meaningfully and combined when possible.
> 2. Interpretability and diagnosis: If we only provided the end-to-end metric, we would face the "Black Box" problem common in VLN benchmarks—knowing that an agent failed, but not why. By providing modular metrics, ERNav allows for error attribution.
>
> Regarding unified metrics, the Success Rate in EnvNav already functions as the end-to-end metric. Methods lacking certain capabilities use standardized representations generated from our exploration stage, ensuring fair comparison. And methods capable of the full pipeline are evaluated fully end-to-end.
>
> Therefore, ERNav is "End-to-End" in execution capability but "Modular" in evaluation design to support scientific diagnosis. We will revise the text to distinguish them to avoid confusion.

---

> ### Author Response · Authors · 2025-11-22
> **Response to Reviewer HHyV - Part 2**
>
> > ### Weakness 2. Support for methods without exploration
>
> We thank the reviewer for this insightful and constructive suggestion, and we apologize for any lack of clarity in our initial description of the pipeline's flexibility.
>
> (1) **Clarification: Our Design Already Provides a Default Trajectory Source.**
> The reviewer correctly points out the need for a default trajectory generator to allow representation-only methods (like ConceptGraphs) to participate in the full pipeline evaluation. However, our existing design already fulfills this role by utilizing our Dual-Sighted Strategy (DSS)—an advanced, high-efficiency exploration module—to generate the required trajectory. As noted in Lines 237–238, for all baselines that do not include their own exploration module, we use this single, high-quality DSS trajectory as the source for their representation construction.
>
> (2) **Rationale for Using the Best Trajectory (Fairness and Upper Bound).**
> We deliberately chose to use the superior DSS trajectory rather than a simpler, lower-performing rule-based one to evaluate representation baselines for two critical reasons:
> 1. Maximizing Fairness in Perception. Our primary goal in evaluating EnvRep methods is to isolate the performance of their representation learning. By providing them with the highest-quality trajectory, we ensure that any performance difference is due to the representation architecture itself, not a bottleneck imposed by poor exploration.
> 2. Establishing the Feasible Upper Bound. Using a suboptimal exploration module would artificially degrade the performance of all representation baselines. By using DSS, we are able to establish the highest feasible performance upper bound for these methods, providing a more meaningful and honest comparison of their inherent representational capabilities.
>
> Therefore, our DSS module serves the intended function of the reviewer’s suggested default generator while ensuring the most robust and fair comparison for the crucial representation stage. We will explicitly clarify in the revised manuscript that the DSS trajectory is the default, high-quality perception source used for all baseline comparisons.
>
> > ### Weakness 3. Human Performance
>
> We appreciate the reviewer’s suggestion. Human performance is partially included in ERNav, but its applicability differs across the three subtasks:
> 1. EnvExp: Table 1 in the paper already reports human exploration results. We note, however, that this is not a strict upper bound, as human trajectories often contain sub-optimal or biased behaviors (e.g., uneven room coverage, preference-driven scanning patterns).
> 2. EnvRep: Measuring human performance is not applicable for this subtask. EnvRep requires the agent to produce a scene representation from raw sensor inputs. This is an intermediate representation task tailored to the cognitive process of an embodied AI system. Humans do not perceive the world in terms of point clouds or semantic scene graph nodes; thus, they cannot realistically execute or be measured on this task.
> 3. EnvNav: The instructions used in EnvNav are derived from existing VLN benchmarks. These original benchmarks already contain Human Success Rate (SR) data, which reflects human capability in following the exact instructions in the environment. We will include these human navigation results in Table 3 of the revised manuscript to serve as a meaningful reference for the final task performance.
>
> > ### Weakness 4. Demonstrations and limitations
>
> We thank the reviewer for the helpful suggestion. ERNav already includes several demonstrations in Figure 2 and Figure 3, such as visualizations of the exploration trajectory, object-level and region-level scene representations, and examples of the final scene-graph–based reasoning. As for limitations and future research directions, these are already discussed in Appendix A.8.  We will add additional qualitative demonstrations in the revised version to further illustrate system behavior.

---

> ### Author Response · Authors · 2025-11-22
> **Response to Reviewer HHyV - Part 3**
>
> > ### Question 1. RGBD noises
>
> Thank you for the question. We apologize for the lack of clarity regarding the source of the noise.
>
> 1. **Source of Noise**
>
> To clarify, ERNav does not artificially introduce synthetic noise to the observations. Instead, the "noise" is inherent to the raw data source we utilize. We use the Matterport3D dataset [1], which consists of 3D meshes reconstructed from real-world scans using commodity RGB-D sensors. By conducting exploration on these real-world scans, our benchmark naturally incorporates two types of realistic sensor noise:
> - Sensor Quantization and Dropouts: The depth maps rendered from these scans retain the artifacts of the original sensors, such as missing depth values (holes) on reflective surfaces (windows, mirrors) and quantization errors at object boundaries.
> - Geometry Imperfections: Unlike synthetic environments with perfect geometry, the underlying meshes in Matterport3D contain reconstruction noise, such as uneven floor surfaces or incomplete object geometries due to occlusion during the original scanning process.
>
> 2. **Ensuring Realistic Real-World Conditions**
>
> The realism of the noise is ensured by the data collection process itself, rather than a simulation algorithm. As noted in the ScanNet paper[2], real-world scanning data is characterized by "noise and occlusion patterns" that are inherently different from synthetic data. Therefore, we do not need to explicitly "add" noise to ensure realism; we simply preserve the raw, imperfect nature of the scanned environments, which serves as a rigorous testbed for real-world deployment. We will clarify this data sourcing and noise definition in the revised version.
>
> [1] Chang, Angel, et al. "Matterport3D: Learning from RGB-D Data in Indoor Environments." 3DV. 2017.
>
> [2] Dai, Angela, et al. "Scannet: Richly-annotated 3d reconstructions of indoor scenes." CVPR. 2017.
>
> > ### Question 2. L1-CD vs. L2-CD
>
> We thank the reviewer for the question. We adopt Chamfer-L1 following established practice in 3D reconstruction and indoor scene completion benchmarks. Chamfer-L1 is widely preferred because:
> - Robustness to outliers: the L1 norm avoids the disproportionate penalization of a few large errors that often arise in incomplete or occluded indoor scans, leading to a more stable and representative geometric score;
> - Better alignment with structural accuracy: prior works have shown that L1 distance correlates better with reconstruction fidelity, such as edge sharpness, planar consistency, and overall scene geometry;
>  - Community convention: recent reconstruction systems—including Atlas [1], SAP [2], and NeRF-based methods [3]—all adopt L1-based Chamfer metrics for evaluating shapes or scenes.
> Following this standard, we report Chamfer-L1 to fairly assess how exploration quality influences downstream reconstruction.
>
> [1] Murez, Zak, et al. "Atlas: End-to-end 3D scene reconstruction from posed images." ECCV, 2020.
>
> [2] Peng, Songyou, et al. "Shape as points: A differentiable Poisson solver." NeurIPS, 2021.
>
> [3] Azinović, Dejan, et al. "Neural RGB-D surface reconstruction." CVPR, 2022.

---

### Official Review · Reviewer_6CyD · 2025-11-01

**Soundness:** 3
**Presentation:** 3
**Contribution:** 2
**Rating:** 2
**Confidence:** 3

**Summary:**

The paper introduces a new end-to-end benchmark where an embodied agent must first explore a building-scale environment, construct a global spatial representation from noisy RGB-D input, and then interpret natural-language instructions to locate targets in a unified pipeline. Prior benchmarks often assume perfect perception or focus only on navigation or question-answering, neglecting the full embodied process of exploration → representation → language-grounded navigation. They present a baseline method ‹3D‑LangNav› that uses a dual-sighted exploration strategy plus an LLM-grounded reasoning component, and show that existing methods struggle on this benchmark, highlighting significant gaps in current embodied-AI capabilities.

**Strengths:**

The authors propose a single pipeline of exploration-representation-navigation stages, which have been explored in part in previous work.
The authors conduct experiments with various baselines for respective stages (e.g., exploration, representation and navigation),

**Weaknesses:**

- It is unclear why we need separate, explicit exploration and representation stages. For example, prior work such as Chaplot et al., 2020a conducts the exploration, representation and navigation simultaneously. What are the benefits of building a map through pre-exploration before navigation, which may require more computational cost compared to prior work?
  - In addition, most of the building blocks come from existing work (Sec. 3.3), raising a novelty concern about the proposed framework. The main contribution may come from the motivation of it, but this is not well described and supported by numbers.
- The paper mentions that the proposed method takes noisy RGBD observations, but it is unclear which part is noisy (e.g., color, depth, camera poses if applicable, ...). It lacks a description.
  - In addition, how much sensitive is the proposed method to the level of noisy? No quantitative analyses of this is conducted.
- In Table 1, the comparison is not fair as the proposed approach uses exploration steps far more than the baselines. The result with the fixed exploration cost should be provided.
- In Table 2, the proposed model underperforms the baselines in several metrics (e.g., R@1%, MR and MRR in Object-level), but the relevant discussion is little to no provided.
- In Table 3, GridMM shows better (NE) or competitive (SR) performances compared to the proposed model without exploration and representation stages, raising a concern of the necessity of the stages.
- (Minor) The paper structure could be improved by avoiding single, long paragraphs and inappropriate capitalization in the title.

**Questions:**

- Does the proposed agent generalize to representations obtained from different camera configurations, such as different FOV, heights, etc.?

---

> ### Author Response · Authors · 2025-11-22
> **Response to Reviewer 6CyD - Part 1**
>
> We appreciate Reviewer 6CyD for the time and effort in reviewing our paper and offering constructive feedback. Please find our responses to the comments below.
>
> ---
> > ### Weakness 1.1. The necessity of explicit exploration and representation stages
>
> We thank the reviewer for the insightful question. Separating exploration and representation is not only a design choice but a requirement imposed by realistic embodied settings. We clarify the benefits regarding computational cost and system design below:
>
> **(1) Alignment with real-world robotic pipelines (Map-and-Plan).**
> A wide range of deployed embodied systems, from classical SLAM pipelines (e.g., ORB-SLAM2[1]) to modern mobile manipulators (e.g., SayCan[2] and HomeRobot[3]), follow a clear “explore → build map → execute task” workflow. This separation allows for the construction of high-quality, global semantic maps that support complex, long-horizon reasoning, which is often difficult to achieve with the transient, local memories used in simultaneous exploration-navigation methods. ERNav is designed to fill this missing evaluation capability, which existing benchmarks overlook.
>
> **(2) Efficiency: Amortizing exploration cost via a "One-to-Many" paradigm.**
> The reviewer raises a valid concern about the cost of pre-exploration. However, we argue that separate exploration is **strictly more efficient** in realistic, long-term deployments. Joint exploration–navigation approaches (e.g., Activa Neural SLAM[4] and SGNav[5]) operate in a one-to-one fashion. For each new instruction, the agent must re-explore the environment from scratch. This results in high redundancy and computational waste. In contrast, ERNav enables a one-to-many paradigm, where a single pre-exploration can serve all subsequent instructions in that building. The "cost" of exploration is amortized over thousands of downstream tasks. Furthermore, an explicit global map allows classical shortest-path planning, further improving inference efficiency. Thus, while pre-exploration adds an initial setup phase, it significantly reduces the per-instruction inference cost and latency, which is the critical metric for real-time interaction.
>
> **(3) Interpretability and error diagnosis.**
> End-to-end VLN agents behave as black boxes, where complex capabilities are compressed into a binary success/failure metric. For instance, when an agent fails to "find the stove in the kitchen," it remains unclear whether the failure stems from never visiting the kitchen (exploration failure), seeing but failing to recognize the stove (representation failure), or failing to ground the text to the map (reasoning failure). ERNav addresses this by enforcing a multi-step evaluation paradigm analogous to Chain-of-Thought (CoT) analysis in LLMs, which isolates the exact stage of failure (as shown in Tables 1, 2, and 3). Crucially, this modularity serves a dual purpose: it not only provides detailed diagnostic results but also establishes granular targets for optimization. Instead of blindly tuning an end-to-end model, researchers can specifically diagnose bottlenecks and optimize the Exploration Policy for coverage, the Perception Module for fidelity, or the Planner for reasoning. This structured approach transforms the evaluation from a simple performance check into a mechanism that provides clear, actionable directions for future research and system improvement.
>
> **(4) Compatibility: ERNav supports both paradigms.**
> We want to make a key clarification here that we do not assume that all methods must separate exploration and navigation, nor claim it is the only valid paradigm.
> Instead, considering the facts that real-world systems frequently use this pipeline and existing benchmarks cannot evaluate such systems, we design ERNav to fill this missing evaluation capability.
> ERNav remains fully compatible with methods that perform one stage (e.g., exploration-only or reconstruction-only models), two stages jointly, or all three in an end-to-end manner.
> Providing such comprehensive coverage allows map-and-plan methods and end-to-end methods to be compared under a unified evaluation protocol, revealing strengths and limitations that existing benchmarks cannot expose.
>
> [1] Mur-Artal, Raul, et al. "Orb-slam2: An open-source slam system for monocular, stereo, and rgb-d cameras." IEEE transactions on robotics. 2017.
>
> [2] Brohan, Anthony, et al. "Do as i can, not as i say: Grounding language in robotic affordances." CoRL. 2023.
>
> [3] Yenamandra, Sriram, et al. "HomeRobot: Open-Vocabulary Mobile Manipulation." CoRL. 2023.
>
> [4] Chaplot, Devendra Singh, et al. "Learning To Explore Using Active Neural SLAM." ICLR. 2020.
>
> [5] Yin, Hang, et al. "Sg-nav: Online 3d scene graph prompting for llm-based zero-shot object navigation." NeurIPS. 2024.

---

> ### Author Response · Authors · 2025-11-22
> **Response to Reviewer 6CyD - Part 2**
>
> > ### Weakness 1.2. Out Novelty
>
> We would like to clarify the reviewer’s concerns about novelty. We apologize if the novelty and contributions were not highlighted clearly enough in the introduction, where we didn't provide an explicit bulleted list due to page limits.
>
> To address the concern regarding novelty, we summarize our core contributions below:
> 1. **The First Unified Benchmark for Realistic Embodied Navigation.**
> ERNav is the first benchmark that unifies exploration, global 3D representation, and language-conditioned navigation in realistic building-scale environments. To the best of our knowledge, no prior benchmark jointly evaluates all three capabilities under a single, coherent evaluation protocol that reflects real-world robotic workflows. This unified setting itself enables entirely new research questions that current VLN benchmarks cannot support.
> 2. **Novel Evaluation Metrics and Task Designs.**
> Each of the three ERNav subtasks introduces novel evaluation designs:
>     - EnvExp: we propose ObjCov, a novel object-coverage metric that assesses whether the agent has gathered sufficient object-relevant observations for reconstruction, something not captured by prior metrics.
>     - EnvRep: We introduce an auxiliary task to evaluate the 3D representation quality for navigation with corresponding evaluation data.
>     - EnvNav: we are the first to interpret VLN as a 3D scene understanding problem, allowing modern 3D-MLLMs to participate in language-grounded navigation, which is not possible in any existing VLN benchmark.
> 3. **3D-LangNav: A Strong Method with Technical Novelty.**
> We propose 3D-LangNav, which introduces specific technical novelties to solve this pipeline:
>     (i) Dual-Sighted Exploration: A new strategy balancing local reconstruction detail and global context.
>     (ii) Bi-level Scene-Graph Matching: A hierarchical grounding mechanism that reduces reasoning complexity.
>     (iii) LLM-based Reasoning: A fine-tuned LLM for enhanced 3D spatial reasoning.
> Together, these enable a full three-stage end-to-end pipeline and achieve strong performance across the benchmark, establishing a strong baseline for the community.
> 4. **Systematic Cross-Paradigm Analysis.**
> We provide the first systematic comparison across three distinct paradigms: VLN agents, 3D Representation methods, and 3D-MLLMs. This cross-paradigm evaluation reveals where each class of method succeeds or fails and provides actionable insights for future research in complex indoor environments.
>
> We believe these contributions are essential for enabling realistic embodied-AI evaluation and for stimulating new research directions. We acknowledge that the current Section 3.3 may not clearly highlight these novel elements, and we will revise both the introduction and method sections to emphasize the contributions more explicitly.
>
> > ### Weakness 2. Noises in RGBD observations
>
> Thank you for the constructive comment. We appreciate the opportunity to clarify the noise sources and the robustness of our method.
>
> 1. **Clarification on "Noisy RGB-D Observations"**
>
> By "noisy observations," we refer to the inherent imperfections and artifacts present in the raw sensor data derived from real-world scans (e.g., Matterport3D [1]), rather than synthetic noise artificially added to clean data. This distinction is well-established in 3D scene understanding literature. For instance, ScanNet explicitly states that real scanning data is characterized by "noise and occlusion patterns" that are inherently different from synthetic data [2].
> Specifically, we clarify the noise sources regarding the reviewer's question:
>  - Depth & RGB (major noise source): Commodity RGB-D sensors inherently suffer from distance-dependent quantization errors, sensor noise at object boundaries, and "dropouts" (missing depth values) on reflective or transparent surfaces.
>  - Reconstruction Artifacts: Since our agent builds representations from partial egocentric views during exploration (unlike benchmarks that use pre-processed global point clouds), the resulting data naturally contains outliers, floaters, and occlusion-induced incompleteness.
>  - Camera Poses (no noise): In our benchmark setting, we utilize the simulator's sensor poses following Habitat's standard protocols [3]. Thus, the "noise" challenge focuses on perception and reconstruction rather than localization drift.
> We emphasize this to highlight that our method operates under these realistic, imperfect perception conditions, rather than assuming access to idealized, noise-free ground truth.

---

> ### Author Response · Authors · 2025-11-22
> **Response to Reviewer 6CyD - Part 3**
>
> 2. **Robustness to Noise**
>
> While we do not conduct a quantitative sensitivity analysis on synthetic noise levels (as our focus is on realistic data), our framework is explicitly designed to be insensitive to the inherent noise levels described above. We achieve this robustness through three mechanisms:
> - Multi-view Aggregation: As described in Sec. 4.3, we aggregate visual features from multiple viewpoints. This averages out pixel-level sensor noise and transient artifacts, producing stable landmark representations.
> - Outlier Rejection: We employ density-based clustering (DBSCAN) to explicitly filter out geometric outliers generated during the depth back-projection process.
> - Hierarchical Abstraction: By abstracting the scene into object and region nodes (Sec. 4.3 & 4.4), our 3D-LangNav essentially acts as a low-pass filter, smoothing out high-frequency sensor noise before reasoning.
>
> Our experimental results (Table 1 & 2) effectively demonstrate that our method successfully reconstructs and reasons over these "noisy" real-world scans. We will revise Section 3.2 to explicitly define these noise sources to prevent ambiguity.
>
> [1] Chang, Angel, et al. "Matterport3D: Learning from RGB-D Data in Indoor Environments." 3DV. 2017.
>
> [2] Dai, Angela, et al. "Scannet: Richly-annotated 3d reconstructions of indoor scenes." CVPR. 2017.
>
> [3] Savva, Manolis, et al. "Habitat: A platform for embodied AI research." ICCV. 2019.
>
> > ### Weakness 3. Fairness & Exploration Cost in Table 1
>
> We thank the reviewer for closely examining the experimental results. We acknowledge that our method utilizes more steps than baselines. However, we respectfully disagree that the comparison is unfair. The difference in step count stems from the fundamental difference between 2D coverage and 3D reconstruction objectives. We clarify why the current evaluation is rigorous and why a "fixed cost" comparison is technically infeasible for this specific task:
>
> **(1) Efficiency is already mathematically penalized in our metric (ObjCov).**
> We do not ignore the cost of extra steps. As defined in Eq. (2), our primary metric ObjCov explicitly includes an efficiency penalty term $E = 1 - \sqrt{\frac{S_{\text{actual}}}{S_{\max}}}$. Despite utilizing more steps (which heavily lowers the $E$ term), our method still outperforms baselines in the final ObjCov score (41.0% vs 29.0%). This indicates that the gain in reconstruction quality outweighs the cost of extra steps. Thus, the trade-off is algorithmically justified.
>
> **(2) Distinct Objectives: 2D Coverage vs. 3D Reconstruction**.
> The baselines (e.g., FBE, ANS) are designed for 2D Occupancy Coverage as they stop as soon as they see the free space. In contrast, ERNav evaluates 3D Reconstruction. To reconstruct a 3D object fidelity, an agent must observe it from multiple angles and distances (our core motivation). This physically requires more movement than simply detecting "this space is free." Comparing step counts directly is akin to comparing a "glance" with a "detailed inspection." The higher step count is not a flaw but a necessary requirement for high-quality 3D mapping, which ERNav aims to benchmark. This aligns with standard practice in the exploration literature[1-2], where methods are evaluated at their own convergence point rather than under an arbitrary fixed budget. Therefore, the current comparison is fair and follows established conventions.
>
> **(3) Technical Infeasibility of "Fixed Cost" Comparison.** The reviewer suggests a fixed cost comparison, but this is difficult to implement meaningfully due to the self-terminating nature of exploration algorithms. Baselines like FBE self-terminate when no frontiers remain. Forcing them to run for more steps is undefined—they have nowhere meaningful to go, so their performance would not improve. Cutting our method off at fewer steps would result in partial multi-view loops, unfairly destroying the reconstruction consistency that the algorithm is designed to build. Therefore, the standard practice in exploration literature [1,2] is to allow each policy to run to its own convergence criterion while reporting steps, which is exactly what we have done.
>
> **(4) Real-world Workflow Alignment.** As noted in our general response, in realistic "Map-and-Plan" workflows, the map is built once during idle time. The critical bottleneck is usually map quality (can the robot find the cup later?) rather than mapping duration (did it take 5 or 10 minutes?). By prioritizing reconstruction quality, our method aligns better with the downstream requirements of the ERNav pipeline.
>
> In summary, the evaluation in Table 1 follows established practice, reflects real-world robotic workflows, and provides a fair comparison across methods with different exploration strategies.
>
> [1] Chaplot, Devendra Singh, et al. "Learning To Explore Using Active Neural SLAM." ICLR. 2020.
>
> [2] Li, Yimeng, et al. "Learning-augmented model-based planning for visual exploration." IROS. 2023.

---

> ### Author Response · Authors · 2025-11-22
> **Response to Reviewer 6CyD - Part 4**
>
> > ### Weakness 4. Performance in Table 2.
>
> We thank the reviewer for the detailed observation. We would like to provide a more in-depth discussion regarding the trade-offs in Table 2 and clarify the performance dynamics.
>
> **(1) Correction on MR (Mean Rank) Performance.** We respectfully point out a potential misunderstanding regarding the MR metric. As defined in Line 454, MR represents the Mean Rank of the target, where lower is better. In Table 2, 3D-LangNav achieves an MR of 12.1, which is significantly better (lower) than the strongest baseline, OpenScenes-2D (18.6), and ConceptGraphs (27.5). This indicates that, on average, our method places the ground truth much closer to the top of the list than baselines, demonstrating robust global ranking capabilities. Note that the **GT rows are not baselines**. They rely on oracle ground-truth segmentation unavailable to any real system. Their purpose is solely to isolate the effect of segmentation noise on representation quality. Directly comparing prediction-based methods to GT rows may lead to the incorrect impression that a model “underperforms.”
>
> **(2) The Trade-off: Point-level Precision vs. Object-level Robustness.**
> The reviewer correctly notes that 3D-LangNav is slightly lower in R@1% compared to OpenScenes-2D. This is a structural trade-off:
>     - OpenScenes-2D projects features to individual 3D points. This preserves high-frequency texture details, helping it hit the "exact match" (Top-1) for distinctive visual queries.
>     - 3D-LangNav aggregates features into object segments (averaging CLIP embeddings within a 3D mask). While this averaging acts as a denoising step (improving MR and R@10), it smooths out fine-grained texture details, occasionally causing the model to miss the exact Top-1 slot for texture-heavy queries.
>
> **(3) Why Higher Recall (R@10) is Preferred for Navigation.**
> In the context of the full ERNav pipeline, R@10 and R@20 are the critical metrics, not R@1, for two reasons.
>     (i). Pipeline Logic: The role of EnvRep is to function as a "Retriever" (Candidate Generator) for the downstream LLM. The goal is not to perfectly identify the target in one shot, but to ensure the true target is present within the top-$k$ candidates (Recall@k) passed to the LLM.
>     (ii). Validation: As long as the target is in the candidate set, the LLM can leverage spatial reasoning to identify it. The success of this strategy is validated in Table 3. This confirms that our representation strategy (optimizing for Recall and Mean Rank) is better aligned with the ultimate navigation goal than optimizing for Top-1 precision.
>
> **(4) Dominance in Region-Level Grounding.**
> Finally, we emphasize that building-scale navigation relies heavily on identifying regions. As shown in Table 2 (Right), 3D-LangNav dominates all baselines in region-level metrics. This capability is the main driver of our superior navigation performance.
>
> We will add this technical discussion to Section 5.2.2 in the revision to explain these trade-offs clearly.
>
> > ### Weakness 5. Performance in Table 3.
>
> We thank the reviewer for the detailed comparison. We acknowledge that GridMM achieves strong performance. However, we respectfully disagree that this negates the necessity of the ERNav stages. We argue that our paradigm offers unique structural advantages—Interpretability, Optimality, and Scalability—that GridMM (and standard VLN methods) cannot provide.
>
> **(1) Structural Advantages: Why "Map-and-Plan" is Necessary.**
> While GridMM performs well on single episodes, our separated pipeline enables capabilities critical for real-world deployment that end-to-end methods lack:
>     - Scalable Efficiency: As mentioned in our response to Weakness 1, our "one-to-many" paradigm allows the map to be reused for infinite instructions without re-exploration, whereas GridMM requires redundant re-exploration for every task. Our method achieves comparable SR without requiring the agent to move a single step during the inference phase. This "zero-movement inference" capability is what makes the explicit representation stage necessary for real-world efficiency.
>     - Optimal Execution: Once the target coordinate is predicted (EnvNav), our agent can use classical planners (e.g., A*) on the built map to execute the shortest path. In contrast, VLN agents often meander or backtrack; our paradigm guarantees execution optimality.
>     - Interpretability: VLN agents are "black boxes"—when they fail, it is hard to know why. Our pipeline enables explicit diagnosis: we can pinpoint if the failure comes from poor exploration (EnvExp), bad reconstruction (EnvRep), or reasoning errors (EnvNav), as analyzed in our experiments.

---

> ### Author Response · Authors · 2025-11-22
> **Response to Reviewer 6CyD - Part 5**
>
> **(2) 3D-LangNav as a Pioneer for a New Paradigm.**
> We emphasize that 3D-LangNav is proposed as a strong baseline for a new, unified benchmark, rather than a method purely optimized to beat SOTA on existing metrics. The fact that 3D-LangNav achieves a 50% Success Rate (matching GridMM's 49%)  validates that this "Map-and-Plan" paradigm is a viable and competitive alternative to traditional VLN. Our contribution is proving that we can achieve SOTA-level task completion (SR) without step-by-step navigation inference. This opens a new research direction for embodied AI that prioritizes map reuse and global reasoning, which is valuable regardless of slight differences in Navigation Error (NE).
>
> **(3) Why NE is higher for Coordinate Prediction.**
> The reviewer notes our NE is higher than GridMM. This is an intrinsic property of the Coordinate Prediction task vs. Path Following. VLN agents move continuously. Even if they fail, they often stop "on the way" to the target or explore nearby rooms, keeping the Euclidean distance (NE) relatively low. ERNav agents perform global localization. If the model retrieves the wrong landmark, the NE penalty is massive. However, on the decisive SR metric, 3D-LangNav outperforms GridMM, confirming its effectiveness in correctly identifying targets.
>
> We will incorporate these discussions into the revised manuscript.
>
> > ### Weakness 6. Paper Structure
>
> We thank the reviewer for the helpful suggestion regarding writing style and presentation. We acknowledge that some paragraphs in the current draft are overly long, and we will restructure them into clearer, shorter units to improve readability. We also appreciate the note on capitalization in the title and will revise it to follow standard academic conventions. These refinements will be incorporated in the revised version.
>
> >### Question 1. Generalization to camera configurations
>
> We thank the reviewer for raising this question. Only the exploration stage in our pipeline depends on camera configuration. The subsequent representation and navigation stages operate purely over the reconstructed 3D map and RGB-D observations. To directly evaluate the influence of camera configuration, we conducted an additional experiment varying the exploration camera parameters along two axes: HFOV: {60°, 90°, 120°}; Camera height: {1.00m, 1.25m, 1.50m}, while keeping all other components unchanged.
>
> **Table 1: The influence of HFOV and heights.**
>
> | Setting              | EnvExp (Coverage %) | EnvExp (ObjCov %)    | EnvExp (F1 %) | EnvRep (R@20 %) | EnvRep (MR %) | EnvNav (SR %) |
> |----------------------|----------------------|---------------------|---------------|------------------|---------------|-------------|
> | Default (90°, 1.25m) | 99.4                 | 41.0                | 52.9          | 84.2             | 12.1          | 50         |
> | HFOV = 60°           | 99.4                 | 39.6                | 52.1          | 82.5             | 12.8          | 48         |
> | HFOV = 120°          | 99.4                 | 38.2                | 53.3          | 83.8             | 12.3          | 49         |
> | Height = 1.00m       | 99.5                 | 40.8                | 23.1          | 71.9             | 16.1          | 37         |
> | Height = 1.50m       | 99.2                 | 40.5                | 54.4          | 84.0             | 12.2          | 50         |
>
> The results in Table 1 demonstrate that our method is highly robust to camera configurations, with downstream performance fluctuating within a narrow margin.
> - Robustness to FOV: Varying the FOV results in negligible fluctuations in final navigation success. This confirms that our multi-view feature aggregation successfully abstracts consistent semantic representations regardless of single-frame width or pixel density.
> - Robustness to sensor height: At 1.00m, we observe a significant drop in geometric F1 and Navigation SR. This is physically expected: a 1.00m camera sits below standard counter height and is blocked by furniture, causing severe occlusion of tabletops and upper spatial structures. The corresponding drop in performance validates that our benchmark correctly penalizes physical invisibility. However, once the camera clears major occlusions (1.25m and 1.50m), performance stabilizes immediately, confirming the method is robust to sensor placement as long as visibility is maintained.
>
> Crucially, this highlights a structural advantage of our pipeline: the Representation and Navigation stages are decoupled from the robot's hardware. In contrast to end-to-end VLN agents, whose navigation performance is tightly overfitted to the specific camera parameters during training, our method allows one robot to explore while a different agent can navigate on the resulting map. This capability ensures that navigation performance is not brittle to robot heterogeneity.
>
> We thank the reviewer for prompting this analysis, and we will include this discussion in the revised manuscript.

---

> ### Author Response · Authors · 2025-11-28
> **Kind Request for Discussion and Reconsideration of the Score**
>
> We sincerely thank Reviewer 6CyD for the valuable feedback and insightful suggestions that have helped us significantly refine our work.
>
> We have posted a detailed response and uploaded a revised manuscript, which includes new experiments and crucial clarifications addressing all the concerns you raised.
>
> We would be very grateful if you could confirm whether our responses and revisions have satisfactorily resolved your initial concerns. Your final input is invaluable for ensuring the quality and clarity of our work, and we stand ready to engage in any further discussion.

---

### Author Response · Authors · 2025-11-22
**Response to All Reviewers**

We thank all reviewers for their valuable feedback and constructive suggestions. We are encouraged by the recognition of the significance of our work in addressing the critical gap between existing benchmarks and real-world embodied AI settings (Reviewers HHyV, zBhR, oc3P). We are pleased that the novelty of our unified pipeline was well received (Reviewers 6CyD, oc3P). We also appreciate the acknowledgment of our comprehensive experimental analysis (Reviewers 6CyD, HHyV, zBhR), as well as the effectiveness of our proposed 3D-LangNav baseline (Reviewers HHyV, oc3P).

We have carefully reviewed all comments and provided detailed responses to each reviewer. A revised manuscript, addressing all feedback and concerns, will be uploaded soon.

We would like to clarify two core design philosophies of ERNav explicitly:
1. **The Rationale for the "Exploration -> Representation -> Navigation" Evaluation Pipeline:**
Separating exploration and representation is a deliberate design choice motivated by interpretability, real-world alignment, and efficiency.
    - Interpretability and Granular Optimization Targets: Conventional VLN evaluations treat the whole process as end-to-end and the model as a "black box", with which complex, compound capabilities are merely measured by a binary status of success or failure. This makes it impossible to determine whether a model excels or fails in exploration, perception/representation, or reasoning, or to determine whether a failure stems from poor exploration, faulty perception, or reasoning errors. In contrast, **we firmly believe that given their complex nature, embodied agents should be evaluated in a multi-step manner similar to Chain-of-Thought (CoT) LLM systems, meaning that multiple steps and tasks should be used in the evaluation to provide a multifaceted understanding of the different aspects of agent capabilities**. We also refer to the step-wise evaluation methods commonly used by CoT evaluations, e.g., CoT-based LLM systems are often evaluated or analyzed step-by-step using the [GSM8K](https://huggingface.co/datasets/openai/gsm8k) benchmark. Guided by that principle, our modular pipeline is intentionally designed to maximize the interpretability of the evaluation results. Crucially, this modularity serves a dual purpose: it not only provides detailed diagnostic results but also establishes granular targets and clear directions for model optimization. Instead of blindly tuning an end-to-end model to maximize a final reward, researchers can specifically diagnose the failures and optimize the Exploration Policy for coverage (EnvExp), the Perception Module for reconstruction (EnvRep), or the Planner for reasoning (EnvNav). As shown in our experiments, this decomposition provides actionable insights into fine-grained agent capabilities that binary metrics can not provide.
    - Real-World Alignment: This "Map-and-Plan" paradigm mirrors deployed embodied systems for long-term operations, which build a map before executing tasks. For example, a holdhouse or a warehouse robot should not operate under the assumption that it will always be in the exploration mode, with which it does not hold information about the environment beyond its immediate observations or its current task session. Instead, if circumstances permit, an embodied agent should be able to perceive and comprehend the environment thoroughly for all future tasks to maximize the efficiency and success rate of task execution. The way ERNav structures the evaluation pipeline explicitly reflects this observation.
    - Efficiency (One-to-Many): This is linked to the second point. Unlike joint exploration-navigation methods that re-explore for every instruction, ERNav enables a one-to-many paradigm. A single exploration pass constructs a global map that serves all subsequent instructions in that environment, significantly reducing redundant computation. For example, if the environment hasn't changed, an embodied agent should not need to explore again to find the same cup.

2. **The Difficulty and Challenges of EnvNav:**
We emphasize that EnvNav is a rigorous and non-trivial reasoning task:
    - High Difficulty: Far from being easy, our experiments show that state-of-the-art 3D-MLLMs achieve less than 20% Success Rate on EnvNav. This confirms that reasoning over building-scale environments with complex spatial constraints remains an open challenge.
    - Scale & Complexity: Unlike room-level benchmarks (e.g., ScanRefer or MSR3D), EnvNav requires reasoning over global coordinates across multiple rooms and floors without ground-truth metadata.
    - Coordinate Grounding: Predicting global coordinates is not redundant; it is the fundamental interface for spatial reasoning. Without map-based grounding, models fail to localize targets effectively.

We hope our responses address all concerns and clarify any misunderstandings regarding our work. Thank you again for your thoughtful reviews and constructive insights.

---

### Author Response · Authors · 2025-11-24
**Revised Manuscript**

Following the constructive suggestions from the reviewers, we have revised our manuscript to address the feedback and improve the quality and clarity of our work. The revised sections are marked in **red text** throughout the manuscript. Below, we summarize the major changes:

### **1. More Experiments and Results**
We introduced additional experiments to strengthen the evaluation, including (i) a camera-configuration robustness analysis (varying HFOV and height), (ii) a controlled stage-dependency study demonstrating that exploration quality directly affects representation and navigation performance, and (iii) a newly added human performance baseline for EnvNav to provide an upper-bound reference.

### **2. More Detailed Descriptions and Analysis**
We added new subsections to clarify implementation details and evaluation protocols (including how representation-based methods and 3D-LLMs are adapted), provided a dataset statistics and data source summary for each subtask, included a discussion on dynamic scenes, a noise analysis clarifying real-world sensing and reconstruction artifacts, and expanded the analysis of performance trade-offs in Tables 1–3.

### **3. Structural Revisions (Introduction & Related Work)**
We revised the Introduction to explicitly describe the critical roles of the exploration and representation stages. We also incorporated a bulleted summary of contributions to clearly present our novelty and the empirical evidence supporting our motivation.
Furthermore, we updated the Related Work section to include a systematic comparison with MSR3D. This revision explicitly delineates differences in scene scale, task formulation, input assumptions, and evaluation protocols, thereby more clearly positioning ERNav within the landscape of existing benchmarks.

We sincerely thank the reviewers for their thoughtful suggestions and constructive feedback. We believe these revisions address the concerns raised and significantly enhance the quality of our manuscript. We hope these changes meet your expectations, and we welcome any additional feedback to further refine our work.

---

### Meta-Review · Area_Chair_3BhZ · 2026-01-08

**Summary:**

This work proposes a unified and realistic benchmark for embodied AI with exploration, representation, and navigation. It receives mixed reviews: 1 borderline accept, 2 borderline rejects, and 1 clear reject. Reviewers raised concerns about benchmark and task setting, insufficient comparisons and analysis in experiments, and lack of discussion on related works. The most critical concern is the benchmark and task setting, which is challenged by most reviewers from multiple perspectives.

**Reviewer Concerns:**

Some concerns are addressed by adding more explanations and experimental results in the rebuttal. However, not all concerns are fully resolved, especially the common concern on task and benchmark settings.

**Reviewer Scores:**

It is likely that reviewers will keep a borderline score, leaning towards rejection.

---

### Decision · Program_Chairs · 2026-01-26

Reject